# Heparan sulfates are critical regulators of the inhibitory megakaryocyte-platelet receptor G6b-B

Timo Vögtle[1], Sumana Sharma[2], Jun Mori[1], Zoltan Nagy[1], Daniela Semeniak[3], Cyril Scandola[4], Mitchell J Geer[1], Christopher W Smith[1], Jordan Lane[5], Scott Pollack[5], Riitta Lassila[6,7], Annukka Jouppila[8], Alastair J Barr[9], Derek J Ogg[10], Tina D Howard[10], Helen J McMiken[10], Juli Warwicker[10], Catherine Geh[10], Rachel Rowlinson[10], W Mark Abbott[10], Anita Eckly[4], Harald Schulze[3], Gavin J Wright[2], Alexandra Mazharian[1], Klaus Fütterer[11], Sundaresan Rajesh[12], Michael R Douglas[13,14,15], Yotis A Senis[1,4]*

[1]Institute of Cardiovascular Sciences, College of Medical and Dental Sciences, University of Birmingham, Birmingham, United Kingdom; [2]Cell Surface Signalling Laboratory, Wellcome Trust Sanger Institute, Cambridge, United Kingdom; [3]Institute of Experimental Biomedicine, University Hospital Würzburg, Würzburg, Germany; [4]Université de Strasbourg, Institut National de la Santé et de la Recherche Médicale, Etablissement Français du Sang Grand Est, Unité Mixte de Recherche-S 1255, Fédération de Médecine Translationnelle de Strasbourg, Strasbourg, France; [5]Sygnature Discovery Limited, Nottingham, United Kingdom; [6]Coagulation Disorders Unit, Department of Hematology, Comprehensive Cancer Center, University of Helsinki, Helsinki University Hospital, Helsinki, Finland; [7]Aplagon Oy, Helsinki, Finland; [8]Coagulation Disorders Unit, Helsinki University Hospital Research Institute, Helsinki, Finland; [9]Department of Biomedical Science, Faculty of Science & Technology, University of Westminster, London, United Kingdom; [10]Peak Proteins Limited, Alderley Park, Cheshire, United Kingdom; [11]School of Biosciences, College of Life and Environmental Sciences, University of Birmingham, Birmingham, United Kingdom; [12]Institute of Cancer and Genomic Sciences, College of Medical and Dental Sciences, University of Birmingham, Birmingham, United Kingdom; [13]Institute of Inflammation and Ageing, College of Medical and Dental Sciences, University of Birmingham, Birmingham, United Kingdom; [14]Department of Neurology, Dudley Group NHS Foundation Trust, Dudley, United Kingdom; [15]School of Life and Health Sciences, Aston University, Birmingham, United Kingdom

*For correspondence:
yotis.senis@efs.sante.fr

**Abstract** The immunoreceptor tyrosine-based inhibition motif (ITIM)-containing receptor G6b-B is critical for platelet production and activation. Loss of G6b-B results in severe macrothrombocytopenia, myelofibrosis and aberrant platelet function in mice and humans. Using a combination of immunohistochemistry, affinity chromatography and proteomics, we identified the extracellular matrix heparan sulfate (HS) proteoglycan perlecan as a G6b-B binding partner. Subsequent in vitro biochemical studies and a cell-based genetic screen demonstrated that the interaction is specifically mediated by the HS chains of perlecan. Biophysical analysis revealed that heparin forms a high-affinity complex with G6b-B and mediates dimerization. Using platelets from humans and genetically modified mice, we demonstrate that binding of G6b-B to HS and

multivalent heparin inhibits platelet and megakaryocyte function by inducing downstream signaling via the tyrosine phosphatases Shp1 and Shp2. Our findings provide novel insights into how G6b-B is regulated and contribute to our understanding of the interaction of megakaryocytes and platelets with glycans.

DOI: https://doi.org/10.7554/eLife.46840.001

## Introduction

Platelets are highly reactive anucleated cell fragments, which are produced by megakaryocytes (MKs) in the bone marrow, spleen and lungs. In an intact vasculature, platelets circulate in the blood stream for 7–10 days and are finally cleared by the reticulo-endothelial system in the spleen and liver. Upon vascular injury, however, platelets adhere to the exposed vascular extracellular matrix (ECM), become activated and form a hemostatic plug that seals the wound. Platelet activation must be tightly regulated to avoid hyperactivity and indiscriminate vessel occlusion (*Bye et al., 2016*; *Jackson, 2011*). The mechanisms that inhibit platelet activation include extrinsic factors, such as endothelial-derived nitric oxide and prostacyclin, and intrinsic factors, such as immunoreceptor tyrosine-based inhibition motif (ITIM)-containing receptors (*Coxon et al., 2017*; *Nagy and Smolenski, 2018*).

G6b-B is a unique platelet ITIM-containing receptor that is highly expressed in mature MKs and platelets (*Coxon et al., 2017*; *Senis et al., 2007*). It is a type I transmembrane protein that consists of a single N-glycosylated immunoglobulin-variable (IgV)-like domain in its extracellular region, a single transmembrane domain and a cytoplasmic tail containing an ITIM and an immunoreceptor tyrosine-based switch motif (ITSM). The central tyrosine residues embedded in the consensus sequences of the ITIM ([I/V/L]xYxx[V/L]) and ITSM ([T]xYxx[V/I]) become phosphorylated by Src family kinases (SFKs) and subsequently act as docking sites for the Src homology 2 (SH2) domain-containing protein-tyrosine phosphatases (Shp)1 and 2 (*Mazharian et al., 2012*; *Senis et al., 2007*). The canonical mode of action of ITIM-containing receptors is to position these phosphatases, as well as the SH2 domain-containing inositol polyphosphate 5-phosphatase 1 (SHIP1) in close proximity to ITAM-containing receptors, allowing them to dephosphorylate key components of the ITAM signaling pathway and to attenuate activation signals. The inhibitory function of G6b-B has been demonstrated in a heterologous cell system, by antibody-mediated crosslinking of the receptor in platelets and G6b-B knockout (*KO*) mouse models (*Mazharian et al., 2012*; *Mori et al., 2008*; *Newland et al., 2007*). Findings from these mice demonstrated that the function of G6b-B goes beyond inhibiting signaling from ITAM-containing receptors (*Mazharian et al., 2013*; *Mazharian et al., 2012*). These mice develop a severe macrothrombocytopenia, myelofibrosis, and aberrant megakaryocyte and platelet function, establishing G6b-B as a critical regulator of platelet activation and production. This phenotype was also observed in a G6b-B loss-of-function mouse model (*Mpig6b$^{diYF}$*) in which the tyrosine residues within the ITIM and ITSM were mutated to phenylalanine residues, abrogating the binding of Shp1 and Shp2 to G6b-B and downstream signaling (*Geer et al., 2018*). Moreover, expression of human G6b-B in mouse platelets rescued the phenotype of G6b-B-deficient mice, demonstrating that human and mouse G6b-B exert the same physiological functions (*Hofmann et al., 2018*). Importantly, null and loss-of-function mutations in human G6b-B have been reported to recapitulate key features of the *Mpig6b* KO and loss-of-function mouse phenotypes, including a severe macrothrombocytopenia, MK clusters in the bone marrow and myelofibrosis (*Hofmann et al., 2018*; *Melhem et al., 2016*). Despite the vital role of G6b-B in regulating platelet production and function, its physiological ligand was not known. Although a previous study demonstrated that G6b-B binds to the glycosaminoglycan (GAG) heparin, the functional significance of this interaction was not known (*de Vet et al., 2005*).

Proteoglycans comprise a heterogeneous family of macromolecules, consisting of a core protein and associated unbranched GAG side-chains. Heparan sulfates (HS) are a specific subgroup of GAGs, defined by their basic disaccharide unit. They are structurally related to heparin, which is produced as a macromolecular proteoglycan by tissue-resident mast cells (*Lassila et al., 1997*) and which, following chemical or enzymatic processing, serves as an anti-coagulant (*Chandarajoti et al., 2016*; *Meneghetti et al., 2015*). One of the best studied and abundant HS proteoglycans is perlecan, which is synthesized and secreted by endothelial and smooth muscle cells

into the vessel wall. It is comprised of a large 400-kDa core protein and has three HS chains attached to its N-terminus. A number of proteins reportedly interact with the HS chains and protein core of perlecan, among them are structural components of the ECM, including laminin, collagen IV and fibronectin, and fibroblast growth factor-2 (*Nugent et al., 2000*; *Whitelock et al., 2008*). Of note, the proteolytically released C-terminal fragment of perlecan, called endorepellin, binds to integrin α2β1 and enhances collagen-mediated platelet activation (*Bix et al., 2007*). Perlecan has also been shown to exert anti-thrombotic properties in an ovine vascular graft model through its HS side-chains, although the underlying mechanism has not been defined (*Lord et al., 2009*).

In this study, we identified the physiological ligand of G6b-B, the molecular basis of the G6b-B ligand interactions and the mechanism underlying physiological effects. Our findings demonstrate that G6b-B binds the HS chains of perlecan, as well as to heparin, eliciting functional responses in MKs and platelets. Moreover, we also show that a cross-linked, semisynthetic form of heparin, called anti-platelet anti-coagulant (APAC) (*Lassila and Jouppila, 2014*), beyond inhibiting collagen-mediated platelet aggregation, induces robust phosphorylation and downstream signaling of G6b-B. Collectively, these results reveal that HSs regulate G6b-B signaling and function, providing a novel mechanism by which MK and platelet function is regulated.

## Results

### Identification of perlecan as a ligand of G6b-B

To identify the tissue expressing the physiological ligand of G6b-B, we generated a recombinant mouse G6b-B Fc-fusion protein (mG6b-B-Fc), consisting of the murine G6b-B ectodomain and the human IgG-Fc tail (to mediate dimer formation), which we used to stain frozen mouse tissue sections. We consistently observed prominent staining in large vessels, including the vena cava and aorta, and also in smaller vessels in the liver and spleen, that were not observed with the negative control (IgG-Fc) (*Figure 1*), suggesting the presence of G6b-B ligand in vessel walls. The highly vascularized bone marrow sections showed a more diffuse staining, indicative of the presence of the ligand in the bone marrow ECM (*Figure 1*).

Because of the strong signals and easy accessibility of the vena cava, we incubated vena cava homogenates with mG6b-B-Fc and protein G sepharose beads to precipitate and identify G6b-B binding partners. SDS-PAGE and colloidal coomassie staining revealed bands of high molecular weight that were absent in the negative control (IgG-Fc pulldown, *Figure 1—figure supplement 3*). Bands were excised and proteins identified by mass spectrometry, revealing basal membrane-specific HS proteoglycan (HSPG) core protein or perlecan as the most abundant protein specifically pulled-down with mG6b-B-Fc (*Table 1*).

The interaction with perlecan was verified using an in vitro binding assay, which measured the binding of soluble mG6b-B-Fc to immobilized molecules. mG6b-B-Fc bound robustly to perlecan, but not to bovine serum albumin (BSA) (control) or other ECM molecules, including collagen I and IV, various forms of laminin (111, 411, 421, 511 and 521), fibronectin or the related and recombinantly expressed HSPGs syndecan-2 or agrin (*Figure 2A*). Hence, the laminin and collagen identified by G6b-B pulldown and mass spectrometry (*Table 1*) most probably represented perlecan-associated proteins (*Battaglia et al., 1992*) rather than direct binding partners of G6b-B. Human G6b-B-Fc (huG6b-B-Fc) showed binding characteristics similar to those of mG6b-B-Fc (*Figure 2A*).

Treatment of perlecan with the enzyme heparinase III, which removes the HS side-chains, significantly reduced G6b-B binding to immobilized perlecan (*Figure 2B*), indicating that G6b-B binds to the HS side-chains rather than the protein core. This observation was further supported by a competition assay, in which the addition of soluble HS inhibited the binding of G6b-B to immobilized perlecan (*Figure 2C*). Of note, unfractionated heparin, which is closely related to HS, also interfered with G6b-B binding to perlecan and streptavidin-immobilized biotin-conjugated heparin and also bound directly to G6b-B-Fc (*Figure 2A*).

To gain further insights into the structural requirements of the G6b-B–ligand interaction, we tested heparin oligomers of different lengths (4, 8, 12 and 20 saccharide units, degree of polymerization (dp)4, dp8, dp12 and dp20, respectively) and selectively desulfated heparin molecules for their binding to G6b-B. In a competition assay, only oligomers of at least eight saccharides were able to block binding of G6b-B to heparin-biotin partially, suggesting that this is the minimum length

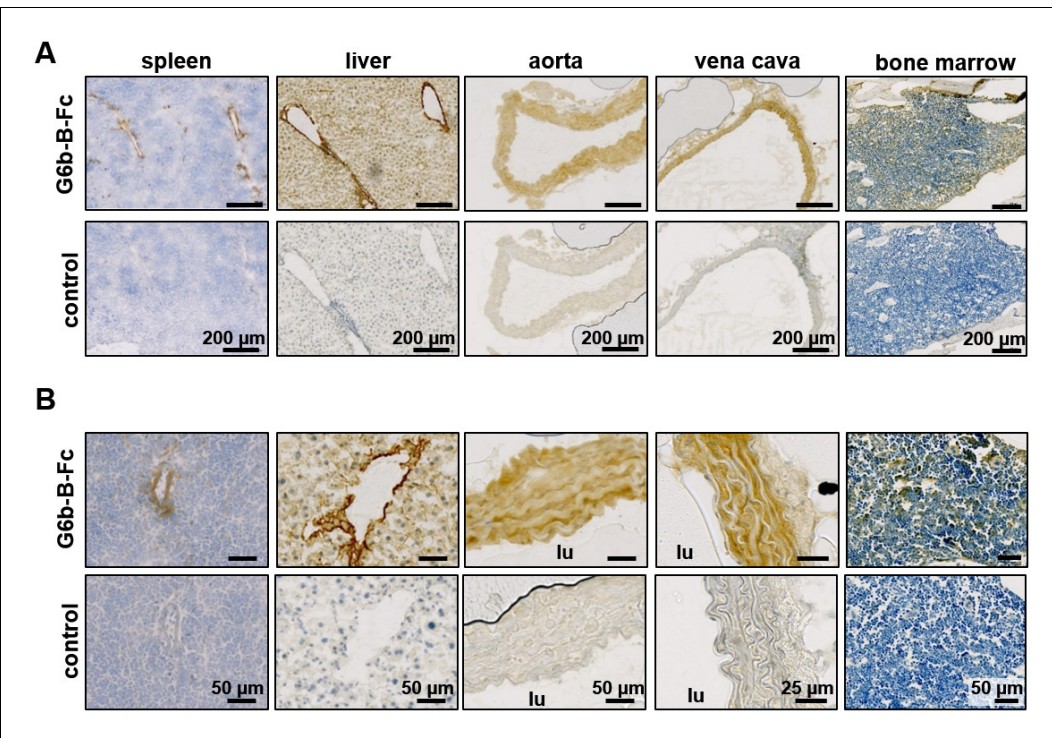

**Figure 1.** Prominent binding of mG6b-B-Fc to the vessel wall. Immunohistochemistry staining of frozen mouse tissue sections with mG6b-B-Fc or human IgG-Fc fragments (control). Bound protein was visualized using a secondary anti-human-Fc-HRP antibody and DAB substrate, prior to counterstaining with hematoxylin. The images were captured by a Zeiss Axio Scan.Z1 slidescanner, and images were exported using the Zeiss Zen software. (**A**) Overview and (**B**) zoomed-in images for the indicated tissues. lu, vessel lumen. Larger overview sections of the tissues are shown in *Figure 1—figure supplements 1*, *2*.

DOI: https://doi.org/10.7554/eLife.46840.002

The following figure supplements are available for figure 1:

**Figure supplement 1.** Overview sections of tissues stained with mG6b-B-Fc or negative control.
DOI: https://doi.org/10.7554/eLife.46840.003
**Figure supplement 2.** Overview sections of tissues stained with mG6b-B-Fc or negative control.
DOI: https://doi.org/10.7554/eLife.46840.004
**Figure supplement 3.** Pull-down of G6b-B binding partners from vena cava lysates.
DOI: https://doi.org/10.7554/eLife.46840.005

required for this interaction (*Figure 2—figure supplement 1A*). In addition, high sulfation of the glycan was found to be important for G6b-B binding, as a loss of occupancy of one sulfation site resulted in a significant drop in the ability of the oligomer to block G6b-B binding to native heparin (*Figure 2—figure supplement 1B*).

As the binding assay results suggested that the G6b-B ligand was primarily composed of HS glycans, we opted to confirm and extend these finding using a genome-scale cell-based CRISPR KO screening approach to identify all of the genes that are required for the synthesis and cell surface display of the G6b-B ligand (*Sharma et al., 2018*). We observed that a highly avid recombinant G6b-B molecule, consisting of the entire ectodomain of biotinylated human G6b-B clustered around phycoerythrin (PE)-conjugated streptavidin, robustly stained several human cell lines, providing the basis for a cellular genetic screen (*Figure 3A*). A genome-wide mutant cell library was generated by transducing Cas9-expressing HEK293 cells with a library of lentiviruses, each encoding a single gRNA from a pool of 90,709 individual gRNAs targeting 18,009 human genes (*Sharma et al., 2018*). Transduced cells that had lost the ability to bind to the recombinant protein were isolated using fluorescent-activated cell sorting, and genes that are required for cell surface binding of G6b-B were identified by comparing the relative abundance of gRNAs in the sorted versus unsorted control

**Table 1.** List of proteins immunoprecipitated with mG6b-B-Fc from vena cava lysates

| Accession number | Name | Peptides | Protein score | Protein score negative control | FE |
|---|---|---|---|---|---|
| E9PZ16 | Basement membrane-specific heparan sulfate proteoglycan core protein (perlecan) | 131 | 607.22 | n.d. | |
| *E9QPE7* | *Myosin-11* | *103* | *468.02* | *719.71* | *0.7* |
| F8VQJ3 | Laminin subunit gamma-1 | 75 | 434.43 | 9.66 | 45.0 |
| *Q5SX39* | *Myosin-4* | *80* | *328.62* | *587.14* | *0.6* |
| *Q8VDD5* | *Myosin-9* | *81* | *318.18* | *513.68* | *0.6* |
| P97927 | Laminin subunit alpha-4 | 56 | 285.20 | n.d. | |
| Q61292 | Laminin subunit beta-2 | 63 | 262.37 | n.d. | |
| *B2RWX0* | *Myosin, heavy polypeptide 1, skeletal muscle, adult* | *61* | *244.66* | *446.14* | *0.5* |
| P02469 | Laminin subunit beta-1 | 57 | 236.87 | n.d. | |
| J3QQ16 | Protein Col6a3 | 61 | 232.99 | 14.76 | 15.8 |
| *G3UW82* | *MCG140437, isoform CRA_d* | *54* | *214.75* | *378.87* | *0.6* |
| *B7FAU9* | *Filamin, alpha* | *58* | *202.67* | *139.51* | *1.5* |
| Q3UHL6 | Putative uncharacterized protein — fibronectin | 48 | 192.76 | n.d. | |
| *Q9JKF1* | *Ras GTPase-activating-like protein IQGAP1* | *31* | *107.79* | *68.57* | *1.6* |
| M0QWP1 | Agrin | 21 | 84.47 | n.d. | |
| P19096 | Fatty acid synthase | 27 | 74.68 | 23.81 | 3.1 |
| Q61001 | Laminin subunit alpha-5 | 23 | 73.24 | n.d. | |
| E9QPX1 | Collagen alpha-1(XVIII) chain | 16 | 59.16 | n.d. | |
| A2AJY2 | Collagen alpha-1(XV) chain | 14 | 53.53 | n.d. | |
| B7ZNH7 | Collagen alpha-1(XIV) chain | 15 | 43.27 | 3.09 | 14.0 |
| *P26039* | *Talin-1* | *11* | *42.29* | *29.93* | *1.4* |

Fold enrichment (FE)=score G6b-B-FC precipitation/score negative control; n.d. = not detectable. Proteins that are prominently present in the negative control (FE < 2) are shown in italic. The protein score was calculated using the SEQUEST HT search algorithm and is the sum of all peptide Xcorr values above the specified score threshold (0.8 + peptide_charge × peptide_relevance_factor where peptide_relevance_factor is a parameter with a default value of 0.4). The full data set, including the mass spectrometry result for the respective band of a G6b-B-FC only sample, is found in **Table 1—source data 1–3**. A picture of a gel and the bands excised for mass-spectrometric analysis are shown in **Figure 1—figure supplement 3**.
DOI: https://doi.org/10.7554/eLife.46840.006

The following source data is available for Table 1:
Source data 1. Mass spectrometry results for proteins precipitated from vena cava lysates with mG6b-B-Fc.
DOI: https://doi.org/10.7554/eLife.46840.007

Source data 2. Mass spectrometry results for proteins precipitated from vena cava lysates with Fc control protein.
DOI: https://doi.org/10.7554/eLife.46840.008

Source data 3. Mass spectrometry results for the proteins detected at the respective height after loading mG6b-B-Fc only (no vena cava lysate).
DOI: https://doi.org/10.7554/eLife.46840.009

populations (*Li et al., 2014*). Using this strategy, we unambiguously identified many genes that are required for HS biosynthesis, beginning with the generation of the tetrasaccharide linkage on the serine residue of the protein backbone (*B3GAT3*, *XYLT2*, *B4GALT7*), the commitment towards the HS pathway (*EXTL3*), HS chain polymerization (*EXT1/2*), and HS chain modification (*NDST1*, *HS2ST1*) (**Figure 3B**). Of particular note, genes encoding the enzymes chondroitin sulfate N-acetylgalactosaminyltransferase 1 and 2 (*CSGALNACT1/2*), which are essential for the

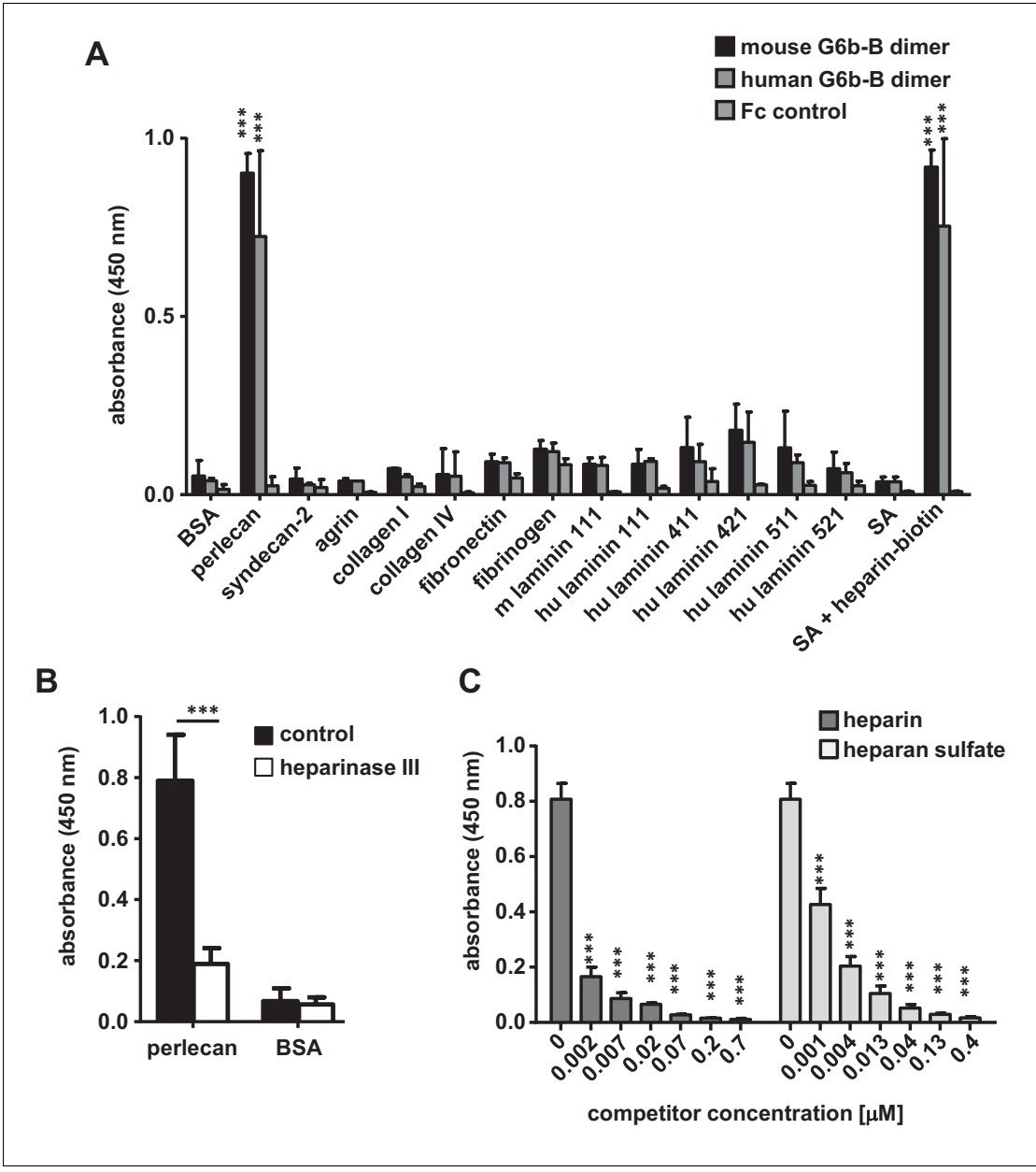

**Figure 2.** G6b-B-Fc binds to heparan sulfate side-chains of perlecan. (A) 96-well plates were coated with the indicated substrates (5 µg/ml) and incubated with mouse G6b-B-Fc (10 µg/ml), human G6b-B-Fc (30 µg/ml) or Fc-control (10 µg/ml). Bound protein was detected with an anti-human-Fc-HRP antibody and 3,3',5,5'-tetramethylbenzidine (TMB) substrate. n = 2–4; SA, streptavidin. (B) Perlecan and bovine serum albumin (BSA) were treated or not with heparinase III (5 mU/ml) prior to blocking, and mG6b-B-Fc binding was measured. n = 4. (C) mG6b-B binding to immobilized perlecan was measured in the presence of the indicated concentrations of heparin and heparan sulfate. n = 3. P-values were calculated using ordinary one-way ANOVA with Dunnett's post-hoc test and asterisks denote statistical significance compared to the respective control. ***, p<0.001. Source files of all binding assays are available in *Figure 2—source data 1*.

DOI: https://doi.org/10.7554/eLife.46840.010

The following source data and figure supplement are available for figure 2:

**Source data 1.** Source data for graphs shown in *Figure 2A–C*.
DOI: https://doi.org/10.7554/eLife.46840.012

**Figure supplement 1.** Loss of heparin sulfation impairs interaction with G6b-B.
DOI: https://doi.org/10.7554/eLife.46840.011

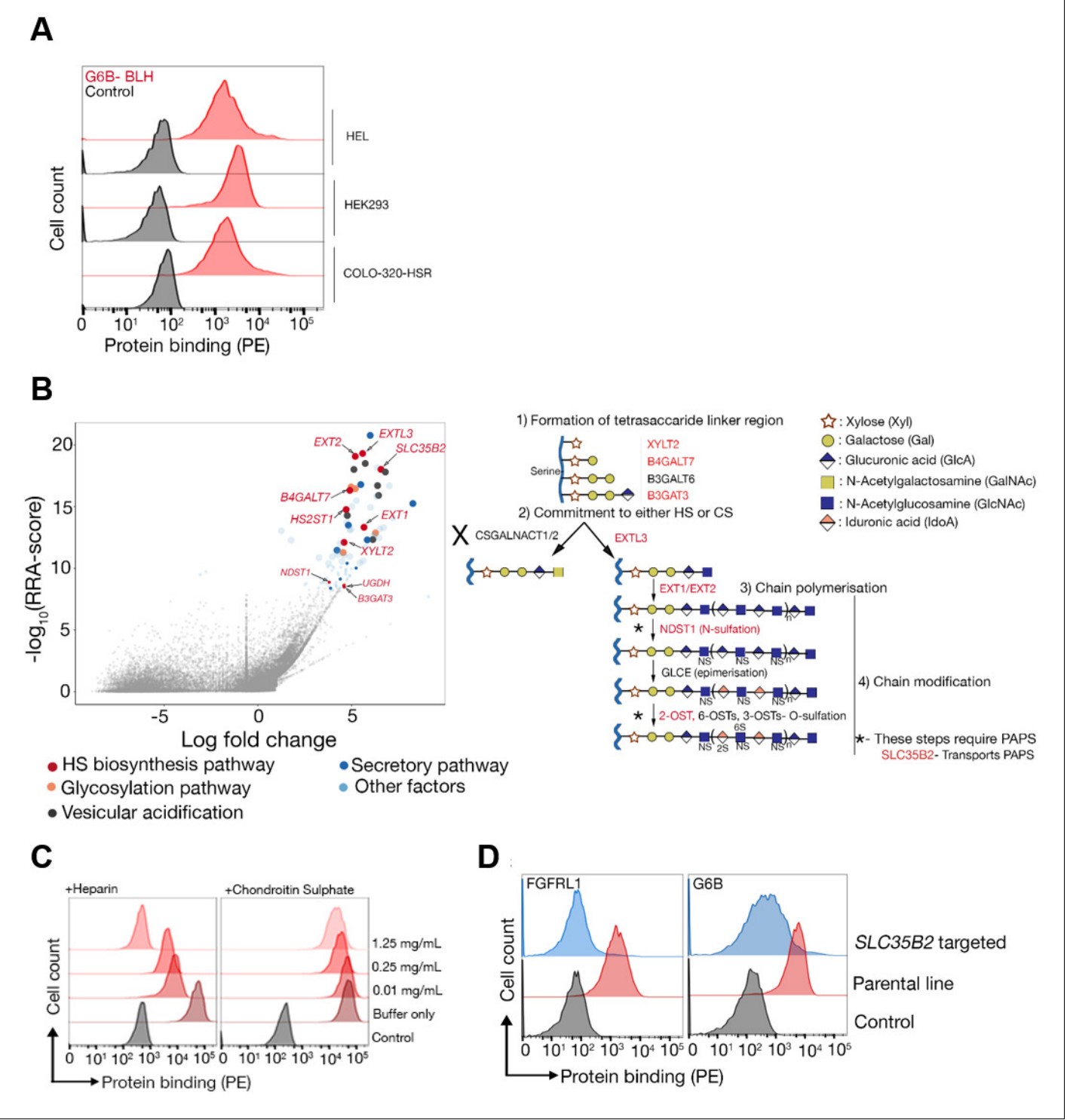

**Figure 3.** The heparan sulfate biosynthesis pathway is required for G6b-B binding to HEK293 cells. (**A**) Recombinant G6b-B, produced as a monomeric biotinylated protein and conjugated to streptavidin-PE to generate an avid probe, binds to HEL, HEK293 and COLO-320-HSR cells. (**B**) A genome-wide loss-of-function approach identifies the HS biosynthesis pathway as the factor required to mediate the binding of recombinant G6b-B to HEK293 cells (left panel). X- and y-axis represent the log-fold-change (LFC) and robust rank aggregation (RRA) score calculated using the MAGeCK software, respectively. Circles represent individual genes and sizes represent the false-discovery rate (FDR): large circle = FDR < 1%, small circle = 1% < FDR < 5%. Genes with FDR < 5% are color coded according to their functional annotation and genes corresponding to the HS biosynthesis pathway are additionally named. The HS biosynthesis pathway is depicted in the right panel with the genes identified in the loss-of-function approach highlighted. Similar results were obtained in HEL cells (not shown). (**C**) G6b-B binding to HEK293 cells was measured by flow cytometry in the presence or absence

*Figure 3 continued on next page*

*Figure 3 continued*

of the indicated concentration of heparin or chondroitin sulfate. One representative out of three experiments is shown. (D) G6b-B loses its binding to cell lines when *SLC35B2*, encoding a transporter required for the sulfation of glycosaminoglycans (GAGs), is targeted. To ensure that the KO cells lack GAGs, a known HS binding protein, FGFRL1, is used as a control that confirms the loss of binding on these cell lines. Source data for the genomic screens in HEK293 and HEL cells are available in *Figure 3—source data files 1–2* and *3–4*, respectively.

DOI: https://doi.org/10.7554/eLife.46840.013

The following source data is available for figure 3:

**Source data 1.** Raw read counts from the screen carried out in HEK293 cells.

DOI: https://doi.org/10.7554/eLife.46840.014

**Source data 2.** MAGeCK output for gene-wise ranking from the screen carried out in HEK293 cells.

DOI: https://doi.org/10.7554/eLife.46840.015

**Source data 3.** Raw read counts from the screen carried out in HEL cells.

DOI: https://doi.org/10.7554/eLife.46840.016

**Source data 4.** MAGeCK output for gene-wise ranking from the screen carried out in HEL cells.

DOI: https://doi.org/10.7554/eLife.46840.017

commitment towards the biosynthesis of chondroitin sulfate chains, were not identified, demonstrating that G6b-B binding to HEK293 cells is mediated by HS, but not by chondroitin sulfate (*Figure 3B*). Moreover, the addition of heparin, but not chondroitin sulfate, inhibited G6b-B binding to HEK293 cells (*Figure 3C*). We also identified *SLC35B2* (Solute Carrier Family 35 Member B2), a gene encoding a transporter protein that translocates 3′-phosphoadenosine-5′-phosphosulfate from the cytosol into the Golgi apparatus, where it is used as a sulfate donor for the sulfation of glycoproteins, proteoglycans, and glycolipids. We validated the involvement of sulfated HSs in mediating G6b-B binding to cells by individually targeting *SLC35B2* and were able to demonstrate that this led to a loss of G6b-B binding relative to the parental cell line (*Figure 3D*). Together, this genetic screen provides further evidence that the physiological ligand of G6b-B is negatively charged HS, corroborating our in vitro binding data.

## Molecular basis of G6b-B interaction with the HS side-chains of perlecan

The extracellular domain of G6b-B is enriched in positively charged residues, especially arginines (12 in 125 amino acids; 9.6% vs 5.6% average frequency in mammalian membrane proteins [*Gaur, 2014*]), which are known to mediate strong binding to heparin (*Margalit et al., 1993*). Prior to obtaining the crystal structure, we generated a structural model of G6b-B using template-based tertiary structure prediction (RaptorX Structure Prediction server) and used this model to aid in the identification of candidate residues for mutagenesis. Examination of the model showed four basic residues (Lys54, Lys58, Arg60 and Arg61) in close spatial proximity to each other on a solvent-exposed loop. We tested whether these amino acids are involved in heparin binding by generating a mutant G6b-B (K54D, K58D, R60E, R61E; *Figure 4—figure supplement 1A*) and by comparing heparin binding to WT G6b-B in transiently transfected CHO cells. An anti-G6b-B monoclonal antibody demonstrated a robust cell surface expression of mutant G6b-B that was comparable to that of WT G6b-B, suggesting that the quadruple mutation did not disrupt protein folding or expression (*Figure 4—figure supplement 1B*). Cells expressing WT G6b-B showed an increase in heparin binding compared to that in non-transfected cells, whereas the cells expressing mutant G6b-B showed impaired binding when compared to WT G6b-B expressing cells, demonstrating that these amino acids (or a subset thereof) are involved in ligand binding (*Figure 4—figure supplement 1C*).

## The crystal structure of the G6b-B extracellular domain (ECD)–dp12–Fab complex

Subsequent to the tertiary structure prediction, we were able to generate crystals of the ternary complex of the ectodomain of G6b-B bound to the heparin oligosaccharide dp12, scaffolded by a G6b-B-specific Fab fragment, and we determined the structure of this complex by X-ray crystallography to 3.1 Å resolution (*Figure 4* and *Table 2*). The construct that was used was N32D, S67A, S68A, S69A, T71A. The N32D mutation was made to remove the single potential N-linked glycosylation

site. Intact mass spectrometry also revealed that after having made the N32D mutation, the measured mass of the protein was 948 Da greater than expected, consistent with O-glycosylation. Subsequent analysis identified five Ser and Thr residues as O-glycosylation sites, of which four were mutated to Ala in successful crystallization experiments.

The solved complex encompasses six protein subunits, a dimer of G6b-B and two Fab fragments. As expected for a Fab-scaffolded structure, crystal packing contacts occur predominantly between the Fab fragment subunits (*Figure 4—figure supplement 2A*), but sparse direct contacts between symmetry-related G6b-B subunits also occur (*Figure 4—figure supplement 2B*).

Confirming the fold of the predicted model, the ectodomain of G6b-B forms an immunoglobulin-like fold of a topology closely resembling the structure of a variable immunoglobulin (Ig) domain (*Figure 4C*) (*Brändén and Tooze, 2009*). A disulfide bond between cysteine residues 35 and 108 (strands B and F, respectively) stabilizes the immunoglobulin (Ig) fold (*Figure 4C*). The backbone does not form the canonical strand C″, and only a very short strand D. In a canonical Ig domain, strand A is part of the sheet formed by strands B–E–D, but in the case of G6b-B, it is part of the opposite sheet (strands C′–C–F–G). The two G6b-B subunits (peptide chains E and F in the coordinate set) superimpose closely relative to the core β-sandwich structure, but divert markedly from each other in the loop connecting strands C′ and D (residues 66 to 81; *Figure 4C*). This loop includes several putative O-glycosylation sites (*Figure 4D*), which were mutated to Ala to ensure homogenous glycosylation of the protein. However, the O-linked glycosylation site Thr73 was retained, and electron density shows the presence of three saccharides attached to Thr73 in both peptide chains (*Figure 4—figure supplement 3*). Although the electron density (resolution 3.1 Å) does not allow the unequivocal identification of the saccharides, the groups could be modeled as galactose, $\alpha$-N-acetyl-D-galactosamine and O-sialic acid, respectively. These glycosyl groups are well separated from the heparin oligosaccharide.

The ectodomain of G6b-B assembles into an apparent dimer with a pseudo two-fold symmetry oriented perpendicular to the extended β-sheet that forms the heparin binding site (*Figure 4C*). Dimer formation of G6b-B is driven by the heparin ligand, as demonstrated by size exclusion chromatography (*Figure 5*). Although G6b ECD was eluted at approximately 12.9 kDa, matching the molecular weight of the monomeric protein, the addition of the heparin oligomer dp12 (3.6 kDa) resulted in a complex of around 30.8 kDa, corresponding to the weight of two G6b-B molecules and one dp12 molecule (*Figure 5*).

The interface between chains E and F buries approximately 800 Å$^2$ of solvent accessible surface area. In line with the modest surface area buried between the two subunits, the interface analysis using the PISA software does not predict a stable complex (*Krissinel and Henrick, 2007*), consistent with the observation that ectodomain dimerization is induced by the heparin ligand. Non-covalent contacts between the two chains consist almost entirely of van der Waals (vdW) and hydrophobic interactions, with Trp65$^F$ and Pro62$^F$ positioned centrally in the interface, contacting Pro62$^E$ and Arg61$^E$, while Trp65$^E$ forms vdW contacts with Val77$^F$. There are very few H-bond interactions (Ser57$^E$-O$\gamma$ – Ala66$^F$-O/Ala68$^F$-N; Lys58$^E$-N$\zeta$ – Arg43$^F$-O) across the interface, and notably the central β-sheet (strands C′–C–F–G–A) is not continuous in that it lacks main chain – main chain hydrogen bonds between the C′ strands of opposing protomers (*Figure 4B*). Nevertheless, dimerization creates a deep cleft, into which the heparin ligand inserts (*Figure 6A*). Crystallization involved a dodeca-saccharide, of which eight residues are visible in the electron density map (*Figure 6—figure supplement 1*), with the central residues 4 and 5 representing sulfated L-iduronic acid (IDS) and D-glucosamine (SGN), respectively. Although the ligand-binding cleft provides partial charge complementarity to the sulfate groups of the heparin ligand (*Figure 6A*), perhaps surprisingly, only one sulfate group (residue SGN5) forms ionic interactions with basic side-chains (SGN5-O2S – Arg60$^F$-N$\varepsilon$ 3.3 Å, SGN5-O3S– Lys109$^F$-N$\zeta$ 3.2 Å, where the superscript refers to the chain ID; *Figure 6B,C*). The other eight polar contacts (within a distance cut-off of 4 Å) involving sulfate groups are with backbone amides (Arg60$^E$, Glu113$^E$, His112$^E$; 2.8–3.3 Å) rather than side-chains, while nine residues, including Lys109$^E$, form vdW interactions with the ligand (*Figure 6B,C*). There is exquisite shape complementarity between the heparin and the surface of the G6b-B dimer, even though the S-shaped ligand only partially fills the ligand-binding cleft.

We next measured the binding affinities of G6b-B for the various ligands using surface plasmon resonance (SPR). The human G6b-B-Fc-His6 homodimer and the human G6b-B-Fc-His6/Fc-StreptagII heterodimer were used as dimeric and monomeric G6b-B molecules, respectively. It is important to

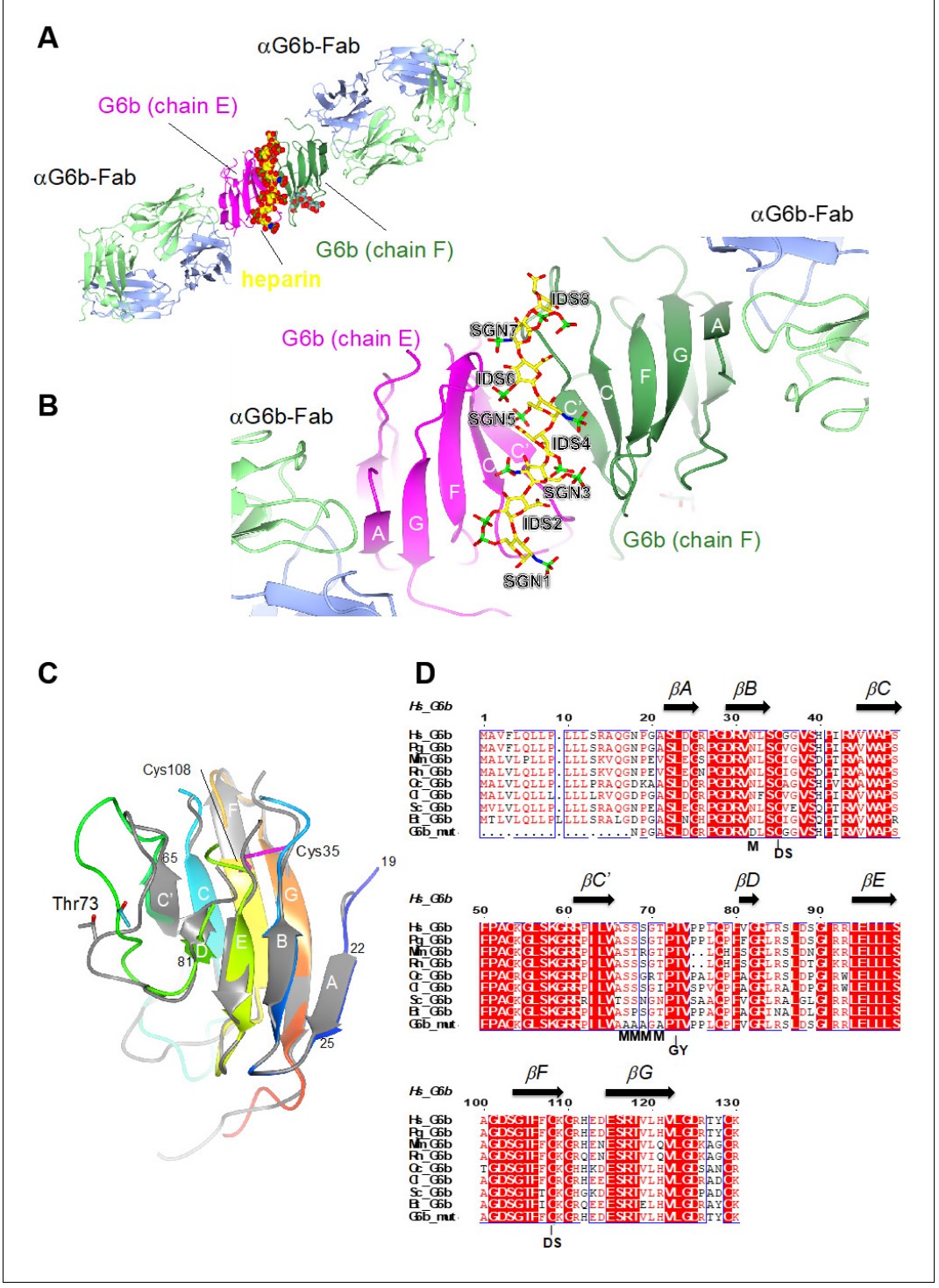

**Figure 4.** Ribbon representation of the ternary complex of the extracellular domain (ECD) of human G6b-B bound to heparin and the Fab fragment of a G6b-B-specific antibody. (**A**) Overview of the structure, with G6b-B colored in magenta and dark green, heparin shown as spheres, and the Fab fragment chains in light green/light blue, respectively. The assembly represents the asymmetric unit of the crystal lattice (space group *C*2). (**B**) Close-up view illustrating the position of the heparin ligand relative to the secondary structure of the G6b-B dimer. Heparin residues (shown as sticks) are sulfated D-glucosamine (SGN) and L-iduronic acid (IDS). The color coding of heparin atoms is: C, yellow; O, red; N, blue; and S, green. β-strands in G6b-B are labeled according to the canonical Ig-fold. (**C**) Superposition of chains F (various colors) and E (gray) of the G6b-B ECD. Strands are labeled according to the canonical β-sandwich topology of the variable Ig domain. The fold of G6b-B deviates from the canonical Ig

*Figure 4 continued on next page*

*Figure 4 continued*
fold in missing strand C'', and as strand A is part of the β-sheet of strands B, E and D. Chain F is color ramped from blue (N-terminus) to red (C-terminus), and the position of the disulfide bond (Cys35–Cys108) is indicated by sticks in magenta. The glycosylation site Thr73 is shown (sticks) with glycosyl groups omitted from the view. (D) Multiple sequence alignment of G6b-B orthologs across mammalian species with secondary structure elements indicated above the sequence. Residue numbers refer to the sequence of human G6b. Conserved residues are boxed, with identities shown as white letters on a red background. Species abbreviations are: Hs, *Homo sapiens*; Pg, *Pan troglodytes* (chimpanzee); Mm, *Mus musculus* (mouse); Rn, *Rattus norvegicus* (rat); Oc, *Oryctolagus cuniculus* (rabbit); Cl, *Canis lupus familiaris* (dog); Sc, *Sos scroftus* (wild boar); and Bt, *Bos taurus* (cattle). G6b_mut is the sequence of the recombinant human G6b-B ECD used in crystallization, with mutations of the five putative glycosylation sites (marked with M). GY indicates the retained O-glycosylation site and DS indicates the disulfide cysteine residues.
DOI: https://doi.org/10.7554/eLife.46840.018
The following figure supplements are available for figure 4:

**Figure supplement 1.** Mutations in G6b-B abolish heparin binding.
DOI: https://doi.org/10.7554/eLife.46840.019
**Figure supplement 2.** Representation of the crystal lattice.
DOI: https://doi.org/10.7554/eLife.46840.020
**Figure supplement 3.** Unbiased σA-weighted difference density map demonstrating the presence of the O-linked glycosyl groups at Thr73.
DOI: https://doi.org/10.7554/eLife.46840.021

note that SPR measures the overall avidity rather than the direct binding affinity of the interactions, factoring in the effects of the multivalent nature of both the receptor (bivalent dimeric form) and the ligands themselves. In the configuration with chip-immobilized G6b-B molecules, heparin bound to both monomeric and dimeric G6b-B with high affinity (low nanomolar range). Similar values were obtained for fractionated (9 kDa) HS and the 12 saccharide heparin oligomer dp12. The binding affinity of perlecan was 366-fold weaker than that of heparin, in the low micromolar range (*Table 3* and *Figure 7A*). The reverse configuration was also tested, in which ligands were biotinylated and immobilized on streptavidin chips. The binding avidity of dimeric G6b-B to perlecan, fractionated HS and heparin was comparable to that measured in the ligand-immobilized configuration (*Table 3* and *Figure 7A*). Interestingly, in the ligand-immobilized configuration, differences in the binding of monomeric and dimeric G6b-B were observed for both heparin and fractionated HS, with the binding of the monomer being approximately 100-fold weaker than that of the dimer (*Table 3* and *Figure 7B*). The apparent decrease in the potency of monomer in the ligand-immobilized configuration versus that in the G6b-B-immobilized configuration is likely to be the result of the ligands themselves being multi-site molecules that are able to bind several sites on the immobilized G6b-B protein surface. Even when the monomeric form of G6b-B is immobilized in the standard assay configuration, the ligands' size and avidity allows them to bind multiple immobilized monomers simultaneously. When the configuration is reversed and the monomeric G6b-B is passed over the flow cell, only a weaker one-to-one binding mode is observed. More efficient binding of the dimeric form in this assay configuration correlates with our crystallography data showing that ligand binding induces dimer formation.

## Biological effects of perlecan, heparin and HS on platelets and MKs

Having established HS as ligand for G6b-B, we examined the effect of surface-bound ligand on platelet function, using an in vitro platelet adhesion assay, in which human platelets were incubated on different substrates and their adhesion was quantified colorimetrically. Platelets bound to fibrinogen, as expected, but failed to adhere to perlecan (*Figure 8A*). However, removal of the HS side-chains by heparinase III treatment resulted in robust adhesion to perlecan. This adhesion might be mediated by interaction of integrin α2β1 with the perlecan protein core (*Bix et al., 2007*), but the contribution of other receptors cannot be excluded. Importantly, perlecan also inhibited the adhesion to fibrinogen and collagen when immobilized together with these substrates. Again, this anti-adhesive effect was abolished upon treatment with heparinase III (*Figure 8A*). These results suggest that the HS side-chains of perlecan negatively regulate platelet adhesion.

**Table 2.** Crystallographic data collection and refinement statistics for the G6b-B ECD–dp12–Fab complex.

**X-ray diffraction data**

| | |
|---|---|
| Beamline | I03, Diamond Light Source |
| Wavelength (Å) | 0.97624 |
| Space group | C2 |
| Cell parameters (Å) | 183.8, 72.34, 131.0, $\beta$ = 124.5° |
| Complexes per asymmetric unit | 1 |
| Resolution range (Å) | 65.27–3.13 |
| High resolution shell (Å) | 3.18–3.13 |
| Rmerge (%)[*] | 17.0 (146.6) |
| Total observations, unique reflections | 74,255/24,543 |
| I/$\sigma$(I)[*] | 4.0 (0.7) |
| Completeness (%)[*] | 97.2 (98.2) |
| Multiplicity[*] | 3.0 (3.1) |
| $CC_{1/2}$[*,†] | 0.991 (0.348) |
| **Refinement** | |
| Resolution range | 63.1–3.13 |
| Unique reflections | 24,543 |
| $R_{cryst}$, $R_{free}$ (%) | 22.6, 26.0 |
| Number of non-H atoms | 7852 |
| RMSD bonds (Å) | 0.01 |
| RMSD angles (°) | 1.18 |
| B-factors | |
| Wilson (Å$^2$) | 77.5 |
| Average overall (Å$^2$) | 84.7 |
| RMSD B-factors (Å$^2$) | 5.737 |
| Ramachandran statistics[‡] | |
| Favored regions (%) | 91.2 |
| Allowed regions (%) | 8.3 |
| Disallowed (%) | 0.5 |

[*] parentheses refer to the high resolution shell.

[†] as defined in **Karplus and Diederichs (2012)**.

[‡] calculated using molprobity (**Williams et al., 2018**).

DOI: https://doi.org/10.7554/eLife.46840.022

To determine whether this inhibitory effect of perlecan on platelet adhesion is mediated via G6b-B, we performed platelet adhesion experiments with platelets from WT and G6b-B knockout (Mpig6b$^{-/-}$) mice (**Figure 8B**). WT mouse platelets exhibited adhesion characteristics that were similar to those of human platelets, with the exception that they adhered weakly to heparinase III-treated perlecan (**Figure 8B**). Importantly, co-coating of fibrinogen together with perlecan reduced the adhesion of WT but not of G6b$^{-/-}$platelets, resulting in enhanced adhesion of Mpig6b$^{-/-}$platelets under this condition. Pre-treatment of perlecan with heparinase III abolished this difference (**Figure 8B**). Adhesion of WT platelets to collagen was inhibited by perlecan in a similar manner as human platelets (data not shown). Platelets from Mpig6b$^{-/-}$could not be meaningfully evaluated on collagen, because of the severe reduction in GPVI surface expression (**Mazharian et al., 2012**). Collectively, these findings demonstrate that the G6b-B–HS interaction inhibited the adhesion of human platelets to the perlecan protein core, collagen and fibrinogen, suggesting an inhibitory effect on integrin and GPVI signaling.

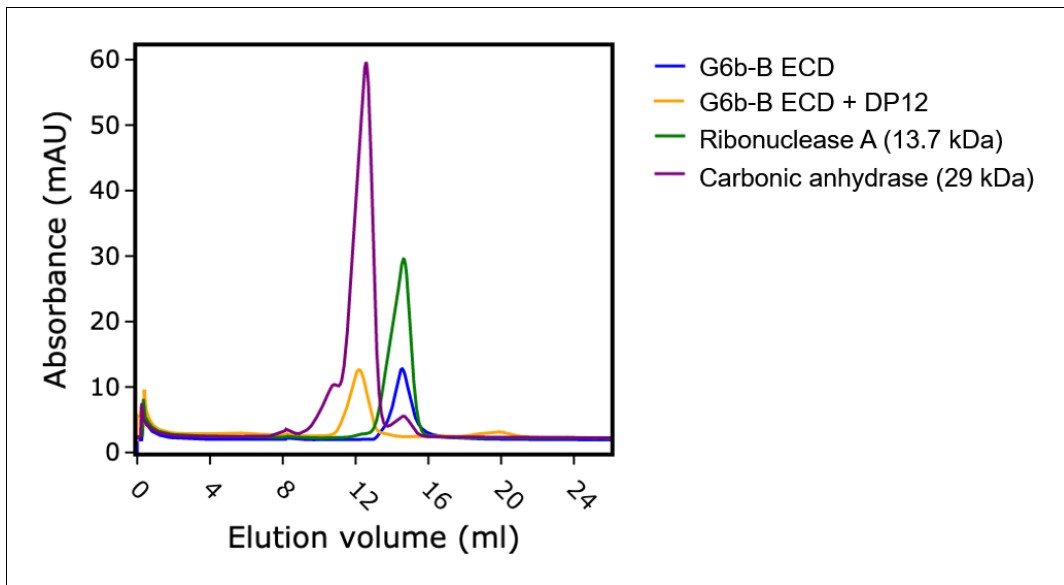

**Figure 5.** Heparin induces G6b-B dimer formation. Size exclusion chromatography of G6b-B ECD. Protein was either analyzed immediately or incubated at 4°C for 1.5 hr in the presence of dp12 before analysis on a Superdex 75 10/300 GL column. Molecular weights were estimated using a calibration curve. Values of 30.8 kDa and 12.9 kDa were obtained for G6b-B ECD in the presence and absence of dp12 (approx. 3.6 kDa), respectively. Ribonuclease A (13.7 kDa) and carbonic anhydrase (29 kDa) are shown for comparison.
DOI: https://doi.org/10.7554/eLife.46840.023

We next investigated the morphological changes in platelets that are adherent to perlecan by microscopy. In contrast to human platelets adhering to fibrinogen, which exhibited characteristic spreading and actin stress fiber formation, platelets adhering to perlecan were small in size and did not spread. The removal of the HS chains of perlecan resulted in a modest increase in size, although the platelets were still much smaller than the platelets adhering to fibrinogen alone (*Figure 8C*).

Mouse WT platelets, like human platelets, did not spread on perlecan and were small (*Figure 8D*), although platelets from *Mpig6b−/−*mice spread to a greater extent, indicating their activation. This was not simply due to the larger size of the *Mpig6b−/−*platelets, as they did not differ in size from WT platelets when spread on fibrinogen, in line with previous findings (*Mazharian et al., 2012*). Moreover, this size difference was abolished upon heparinase III treatment of perlecan, demonstrating that HS also has an activating effect on platelets, presumably through an activation receptor that is inhibited by G6b-B. Of note, platelets from *Mpig6b*diY/F* mice, which express physiological levels of a signaling-incompetent G6b-B, recapitulated the enhanced spreading phenotype of *Mpig6b* KO platelets (data not shown). Hence, we conclude that G6b-B signaling is required to inhibit platelet activation in the presence of HS.

We next investigated the potential effect of perlecan on MKs. Staining of WT mouse bone marrow sections revealed perlecan expression in vessel walls, which co-localized with the sinusoid marker endoglin (CD105) (*Figure 9A* and *Figure 9—figure supplement 1*). This raised the possibility that MK G6b-B is likely to come into direct contact with the HS chains of perlecan in sinusoidal vessels during MK maturation and proplatelet formation. The same observation was made in the bone marrow of G6b-B-deficient animals (*Figure 9A*). Consistent with previous findings (*Mazharian et al., 2012*), however, we observed an increased number of MKs in *Mpig6b−/−*animals (*Figure 9B*), distributed throughout the bone marrow as atypical clusters (*Figure 9C* and *Figure 9—figure supplement 2*). Despite the increase in the number of MKs, *Mpig6b−/−*mice showed similar frequencies of the different maturation stages of MKs, as quantified by EM (*Figure 9D,E*), arguing against an overall defect of MK maturation in *Mpig6b−/−*mice.

To investigate the impact of the G6b-B interaction with HS on MKs, we analyzed the spreading and adhesion of bone marrow-derived MKs on different surfaces in vitro (*Figure 10*). We found that

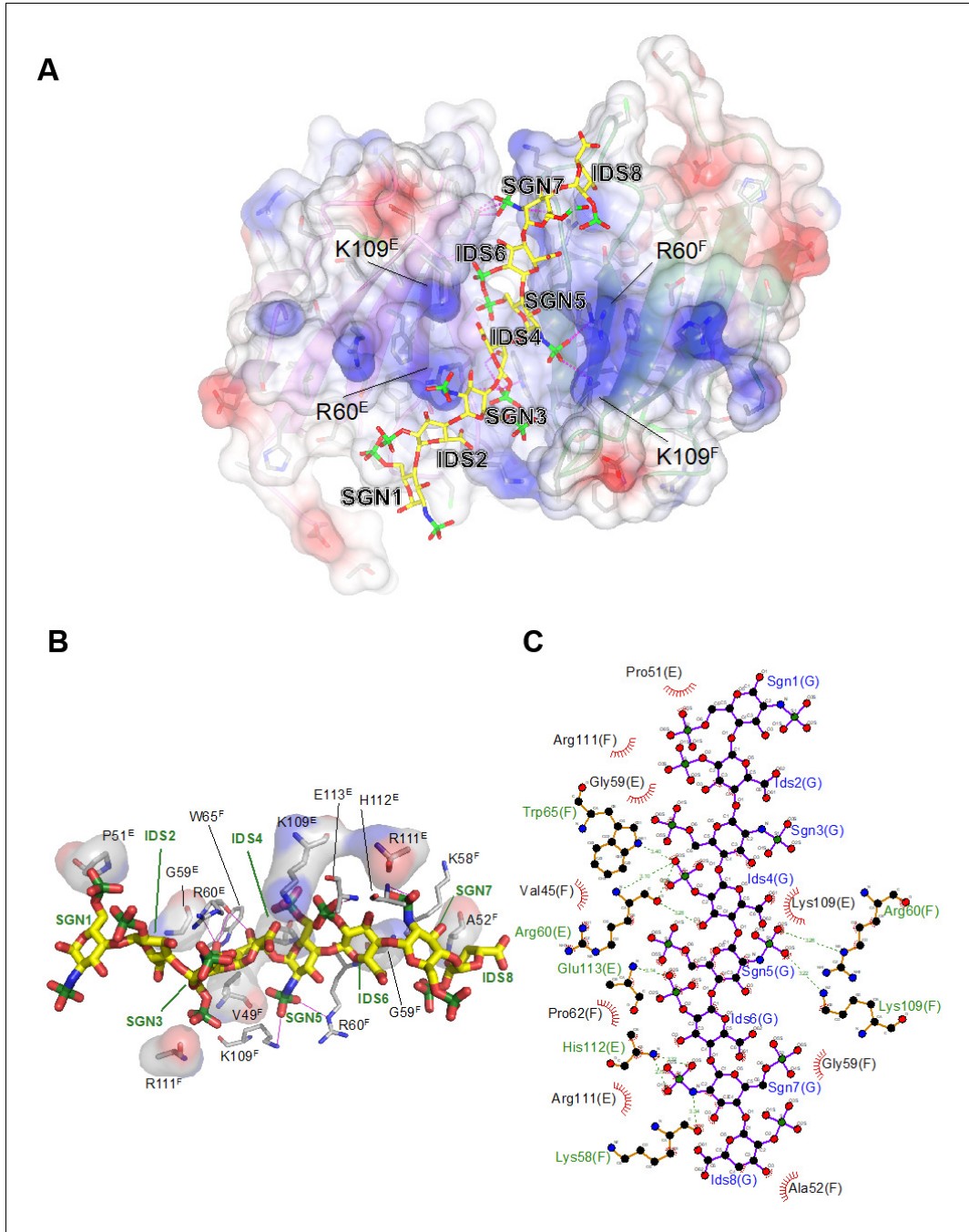

**Figure 6.** Electrostatic surface potential of the G6b-B ECD and representation of non-covalent contacts between heparin and G6b. (A) The G6b-B dimer is shown with a translucent surface colored according to electrostatic surface potential (calculated using CCP4mg). The heparin ligand is shown as a stick model and polar contacts are indicated by dashed lines in magenta. Selected residues are labeled with superscripts indicating the relevant G6b-B protein chain. (B, C) Representation of non-covalent contacts between heparin and the G6b dimer. (B) Residues of G6b-B forming non-covalent contacts with heparin. Polar contacts are indicated by dashed lines in magenta, van der Waals interactions are visualized by showing the relevant residues with their (transparent) molecular surface. Superscript capitals designate the G6b-B protein chain. (C) LigPlot representation of the heparin–G6b-B contacts, with van der Waals or hydrophobic interactions indicated by the bent comb symbol, and polar contacts shown as dashed lines with distance indicated in units of Å.
DOI: https://doi.org/10.7554/eLife.46840.024

The following figure supplement is available for figure 6:

*Figure 6 continued on next page*

*Figure 6 continued*

**Figure supplement 1.** Unbiased $\sigma_A$-weighted difference density map demonstrating the presence of the heparin ligand.

DOI: https://doi.org/10.7554/eLife.46840.025

only very few WT and *Mpig6b*$^{-/-}$MKs adhered to a perlecan-coated surface (*Figure 10*). Whilst perlecan-adherent WT MKs were small in size, *Mpig6b*$^{-/-}$MKs spread to a greater degree on the same substrate. The same effect was observed when perlecan was co-immobilized with fibrinogen, and heparinase III treatment abolished the difference (*Figure 10*). Hence, similar to platelets, exposure of MKs to HS resulted in increased size in the absence of G6b-B, confirming the inhibitory function of this receptor.

We next investigated the biological effects of G6b-B ligands on platelet aggregation in response to collagen, which activates platelets via the ITAM-containing receptor complex GPVI-FcR γ-chain (*Nieswandt and Watson, 2003*). Heparin and HS both enhanced platelet aggregation in response to subthreshold concentrations of collagen (*Figure 11*). This is in line with previous reports and may be explained by binding of these ligands to multiple platelet receptors (*Gao et al., 2011*; *Saba et al., 1984*; *Salzman et al., 1980*), resulting in an overall aggregation-promoting response. We did not find an effect of perlecan on collagen-mediated platelet aggregation at the concentrations tested, suggesting that perlecan must be immobilized to surface in order to provide HS chains at a sufficient density to observe the inhibitory effects observed in adhesion experiments (*Figures 8* and *9*). In addition, multiple direct and indirect effects on platelets through the perlecan protein core, as described previously (*Bix et al., 2007*), may mask an effect of the HS chains in this assay.

To overcome this limitation, we took advantage of the multivalent semisynthetic heparin proteoglycan mimetic APAC (*Lassila and Jouppila, 2014*; *Lassila et al., 1997*) in this assay. APAC consists of unfractionated heparin covalently coupled to a human albumin core, providing a high local density of heparin molecules. In contrast to single-chain heparin, APAC dose-dependently inhibited

**Table 3.** Surface plasmon resonance affinities.

**Immobilized G6b-B receptor (standard configuration)**

| Ligand | G6b-B | $K_{on}$ | $K_{off}$ | $K_D$ (M) |
|---|---|---|---|---|
| Heparin | Monomer | $1.12 \pm 0.39 \times 10^6$ | $2.01 \pm 0.54 \times 10^{-3}$ | $2.00 \pm 1.17 \times 10^{-9}$ |
| | Dimer | $0.60 \pm 0.56 \times 10^6$ | $3.16 \pm 1.17 \times 10^{-3}$ | $7.76 \pm 5.30 \times 10^{-9}$ |
| Fractionated HS | Monomer | $1.33 \pm 0.01 \times 10^5$ | $9.99 \pm 0.16 \times 10^{-4}$ | $7.47 \pm 0.17 \times 10^{-9}$ |
| | Dimer | $1.20 \pm 0.08 \times 10^5$ | $1.71 \pm 1.11 \times 10^{-3}$ | $14.0 \pm 8.26 \times 10^{-9}$ |
| Perlecan | Monomer | $1.94 \pm 1.72 \times 10^2$ | $1.01 \pm 0.37 \times 10^{-4}$ | $7.32 \pm 4.64 \times 10^{-7}$ |
| | Dimer | $5.79 \pm 6.94 \times 10^3$ | $2.28 \pm 2.51 \times 10^{-3}$ | $4.74 \pm 1.34 \times 10^{-7}$ |
| dp12 | Monomer | $0.31 \pm 0.27 \times 10^6$ | $2.39 \pm 1.79 \times 10^{-3}$ | $8.12 \pm 1.22 \times 10^{-9}$ |
| | Dimer | $2.50 \pm 2.72 \times 10^6$ | $4.60 \pm 5.01 \times 10^{-3}$ | $1.84 \pm 0.01 \times 10^{-9}$ |

**Immobilized ligand (reversed configuration)**

| Ligand | G6b-B | $K_{on}$ | $K_{off}$ | $K_D$ (M) |
|---|---|---|---|---|
| Heparin | Monomer | $1.30 \pm 0.29 \times 10^5$ | $8.85 \pm 0.40 \times 10^{-2}$ | $6.99 \pm 1.25 \times 10^{-7}$ |
| | Dimer | $3.28 \pm 0.53 \times 10^5$ | $1.73 \pm 0.04 \times 10^{-3}$ | $5.33 \pm 0.75 \times 10^{-9}$ |
| Fractionated HS | Monomer | $9.22 \pm 2.67 \times 10^3$ | $6.40 \pm 0.33 \times 10^{-3}$ | $7.31 \pm 2.47 \times 10^{-7}$ |
| | Dimer | $3.76 \pm 4.69 \times 10^4$ | $4.58 \pm 6.32 \times 10^{-4}$ | $7.70 \pm 7.21 \times 10^{-9}$ |
| Perlecan | Monomer | $6.73 \pm 3.38 \times 10^3$ | $1.28 \pm 0.24 \times 10^{-3}$ | $2.28 \pm 1.51 \times 10^{-7}$ |
| | Dimer | $4.90 \pm 2.16 \times 10^4$ | $6.78 \pm 2.57 \times 10^{-4}$ | $1.41 \pm 0.09 \times 10^{-8}$ |

Values are means ± SD from two independent experiments.

DOI: https://doi.org/10.7554/eLife.46840.027

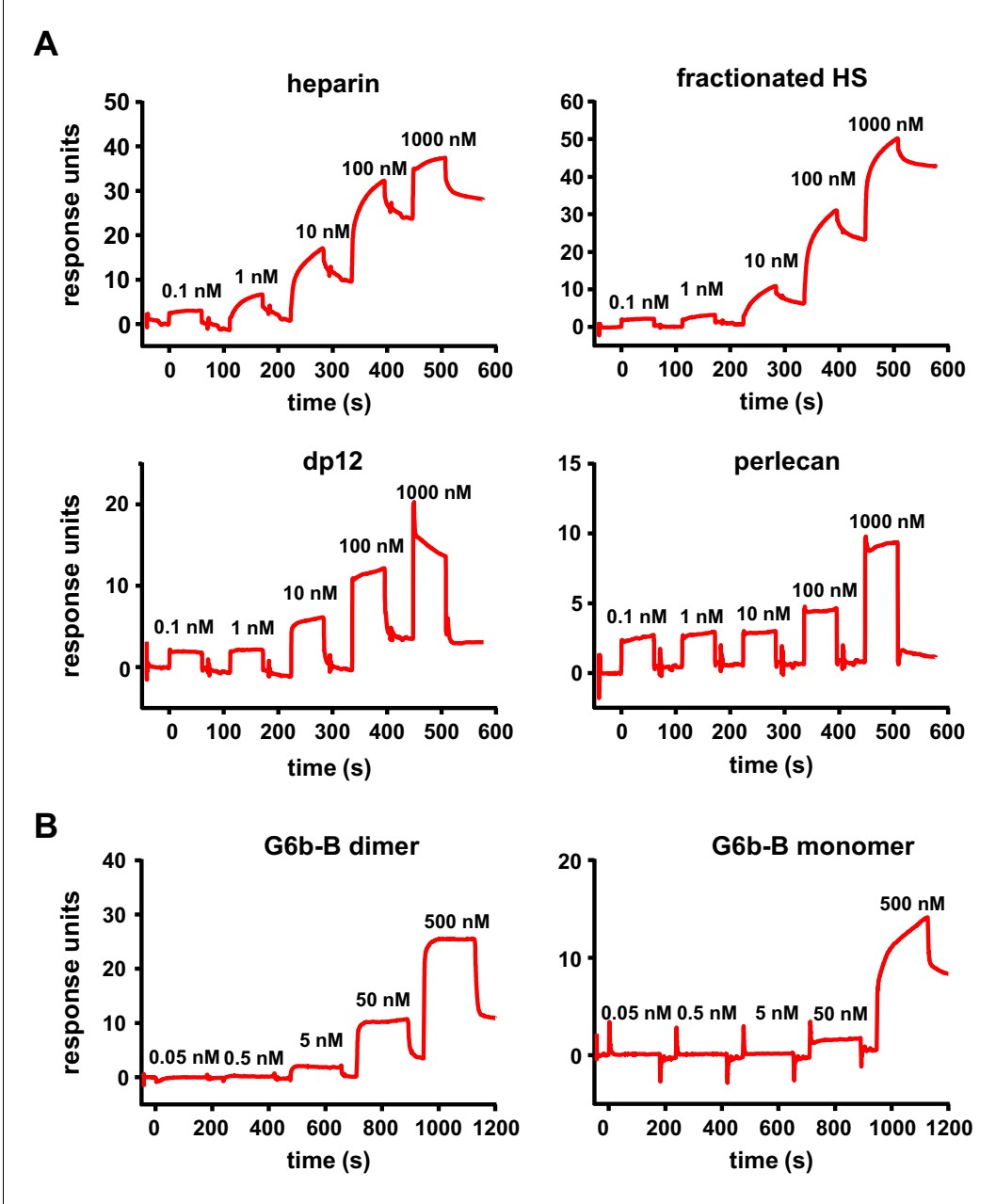

**Figure 7.** High-affinity interaction between G6b-B and its ligands. Representative traces of the surface plasmon resonance experiments, results of which are presented in *Table 3*. (A) Binding of the indicated compound to immobilized dimeric G6b-B in the standard configuration. (B) Results from the reversed configuration, depicting traces of dimeric and monomeric G6b-B binding to immobilized heparin.

DOI: https://doi.org/10.7554/eLife.46840.026

collagen-induced platelet aggregation (*Figure 11*), with an almost complete block observed at 0.5 µM, as previously described (*Lassila and Jouppila, 2014*).

We next examined the effect of heparin and APAC on WT and G6b-B deficient platelets using a flow-cytometric approach, sufficing much smaller sample volumes than aggregation assays, using integrin αIIbβ3 activation (fibrinogen-A488 binding) and degranulation-dependent TLT-1 surface exposure (*Smith et al., 2018*) as markers for platelet activation. APAC and heparin had no detectable effect on WT platelets, but APAC induced robust integrin activation and platelet secretion in G6b-B deficient platelets, demonstrating a platelet-activating effect of this compound in the

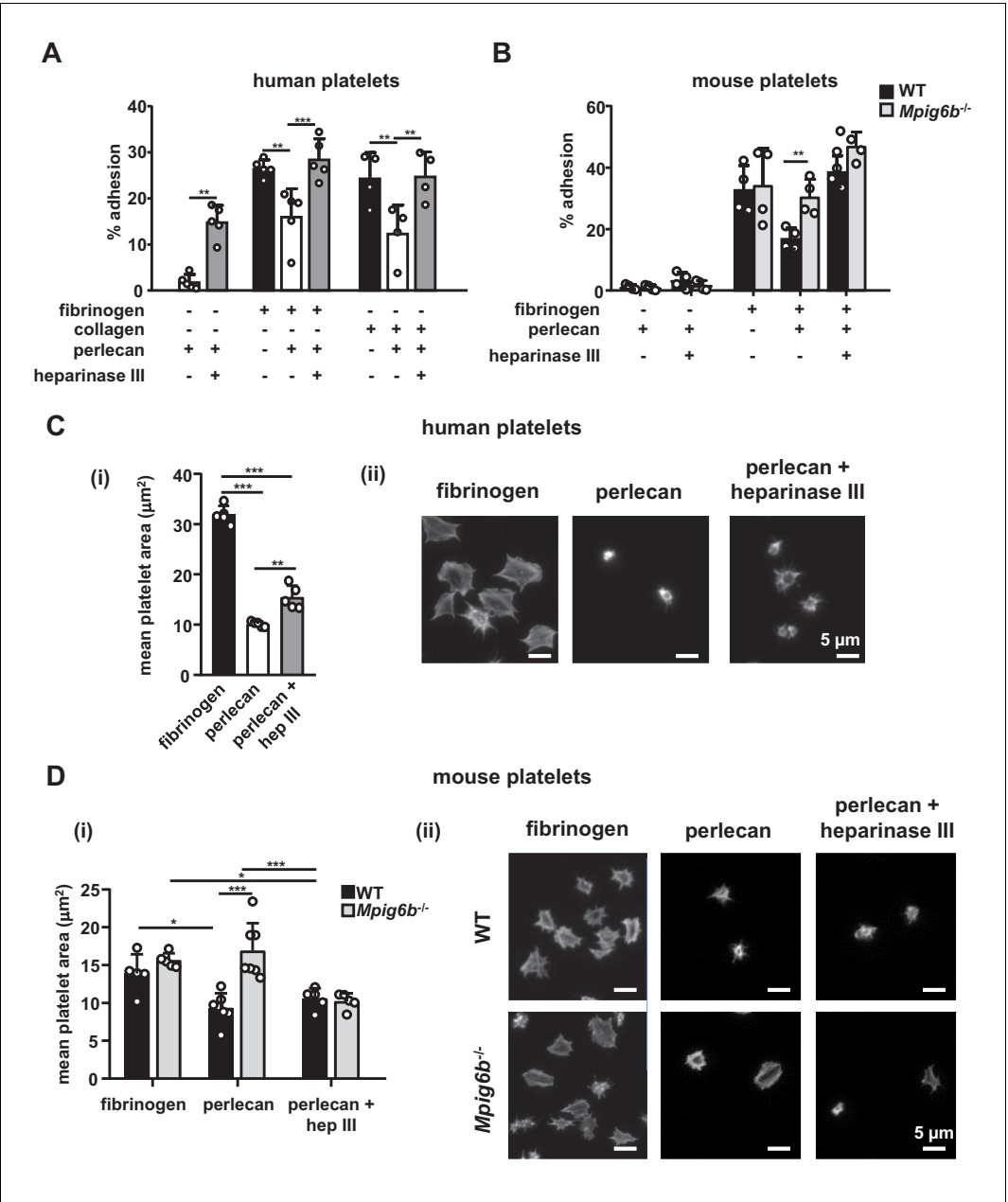

**Figure 8.** Heparan sulfate removal of perlecan facilitates platelet adhesion. The indicated substrates were coated alone or in combination onto wells in 96-well plates (2.5 µg/ml collagen and 10 µg/ml for all other substrates) overnight. Where indicated, wells were treated with 5 mU/ml heparinase III. Platelets from (**A**) humans or (**B**) mice were allowed to adhere for 1 hr and adhesion was quantified colorimetrically with 4-nitrophenyl phosphate(pNPP). (**A**) Human, platelets; n = 4–5 individual donors from 3 to 4 independent experiments; P-values were calculated using one-way ANOVA with Sidak's post-hoc test. (**B**) Mouse platelets; n = 4 samples/condition/genotype from two independent experiments. Owing to severe thrombocytopenia, platelets from up to five mice were pooled for one KO sample. P-values for differences between WT and $Mpig6b^{-/-}$ mice were calculated using two-way ANOVA with Sidak's post-hoc test. (**C, D**) Adhesion of (**C**) human or (**D**) WT and $G6b^{-/-}$ platelets on fibrinogen and perlecan. (**i**) Mean surface area of individual platelets quantified by KNIME software analysis. In panel (**C**) (**i**) n = 5 donors from two independent experiments. P-values were calculated using one-way ANOVA with Sidak's post-hoc test. Total number of cells analyzed: fibrinogen, 1957; perlecan, 239; perlecan + heparinase III, 686.
In panel (**D**) (**i**) n = 5–7 mice/condition/genotype from 2 to 3 independent experiments. P-values were calculated using two-way ANOVA with Sidak's post-hoc test. Total number of cells analyzed: 134–176 for perlecan conditions, and 913–1277 for fibrinogen conditions. *, p<0.05; **, p<0.01; and ***, p<0.001. (**ii**) Representative images of platelets stained for actin with phalloidin-Alexa-488; scale bar: 5 µm; hep III, heparinase III.

*Figure 8 continued on next page*

*Figure 8 continued*

DOI: https://doi.org/10.7554/eLife.46840.028
The following figure supplement is available for figure 8:
**Figure supplement 1.** Mean surface area of individual adherent platelets.
DOI: https://doi.org/10.7554/eLife.46840.029

absence of G6b-B (*Figure 12A*). Next, we aimed to investigate the impact of G6b-B ligands on ITAM-mediated platelet activation in WT and *Mpig6b* KO mice. Owing to severe reduction of GPVI receptor levels in G6b-B deficient animals, we stimulated platelets with an antibody directed against the hemi-ITAM receptor CLEC-2, expression of which is not affected by G6b-B deficiency (*Mazharian et al., 2012*). APAC, but not heparin, significantly inhibited platelet degranulation and fibrinogen binding in response to CLEC-2 stimulation in WT platelets. Importantly, this inhibitory effect of APAC on degranulation was not observed in platelets from G6b-B-deficient animals (*Figure 12B*). Fibrinogen binding was also significantly reduced by APAC in G6b-B-deficient mice, but to a lesser extent than in WT platelets (*Figure 12B*). The inhibitory effect of APAC was also absent in the platelets from *Mpig6b$^{diY/F}$* mice, which express a signaling-incompetent form of G6b-B (*Figure 12C*). Hence, we conclude that APAC suppresses CLEC-2-mediated platelet activation via G6b-B by recruiting the downstream phosphates Shp1 and Shp2. Overall, these findings demonstrate that multivalent G6b-B ligands inhibit platelet activation via (hemi)ITAM receptors, whereas soluble single-chain molecules do not.

## Conjugated heparin induces the phosphorylation of G6b-B and downstream signaling

We performed signaling studies to gain mechanistic insights into the opposing effects of soluble heparin vs. conjugated heparin. Washed human platelets were incubated with heparin or APAC, and their lysates were immunoblotted with an anti-phospho-tyrosine antibody (p-Tyr). Both heparin or APAC induced moderate changes in whole-cell tyrosine phosphorylation as compared to collagen, with APAC having a stronger effect (*Figure 13A*). The most pronounced change observed in response to G6b-B ligation was an increase in the signal intensity of a 150 kDa protein, as well as of a doublet in the heparin- and APAC-treated sample migrating at 27 and 32 kDa, which correlated with glycosylated and non-glycosylated human G6b-B. Hence, we assessed the phosphorylation status of G6b-B using custom phospho-tyrosine-specific G6b-B antibodies directed against phosphorylated ITIM and ITSM of G6b-B (*Figure 13A*), and by immunoprecipitating the receptor and blotting with the p-Tyr antibody (*Figure 13—figure supplement 1A*). Heparin, and to a greater extent APAC, enhanced the basal phosphorylation of G6b-B, which was accompanied by an increase in Shp1 and Shp2 association (*Figure 13A* and *Figure 13—figure supplement 1*). Similar results were obtained with HS, but to a lesser extent than with either heparin or APAC (*Figure 13—figure supplement 1B*). Perlecan did not induce the phosphorylation of G6b-B, in line with our observation in the aggregation assay, suggesting perlecan must be surface-immobilized to have an effect on platelets (*Figure 13—figure supplement 1A*).

Using a quantitative capillary-based gel electrophoresis platform (ProteinSimple Wes), we investigated the effects of heparin and APAC on the phosphorylation status of the tyrosine phosphatases Shp1 (pTyr562) and Shp2 (p-Tyr580 and p-Tyr542), which are essential effectors of G6b-B signaling (*Geer et al., 2018*). Strikingly, APAC induced prominent phosphorylation of Shp1 and Shp2, whereas heparin only induced modest changes in Shp2 phosphorylation (*Figure 13B*). We also observed a marginal increase in SFK phosphorylation (p-Tyr418) in platelets treated with heparin and APAC, correlating with increased phosphorylation of G6b-B under these conditions (*Figure 13B*).

Subsequently, we compared the effects of heparin and APAC on GPVI signaling in response to an intermediate concentration of collagen (3 μg/ml). Although both compounds further enhancing collagen-induced phosphorylation of G6b-B, and although APAC also enhances the phosphorylation of Shp phosphatases (*Figure 13B*), whole-cell phosphorylation remained largely unaltered (*Figure 13A*). Similarly, we also found no inhibitory effect of heparin or APAC on Src (p-Tyr418) and Syk (p-Tyr525/6) phosphorylation, both critical kinases for initiating and propagating GPVI signaling (*Senis et al., 2014*) (*Figure 13B*).

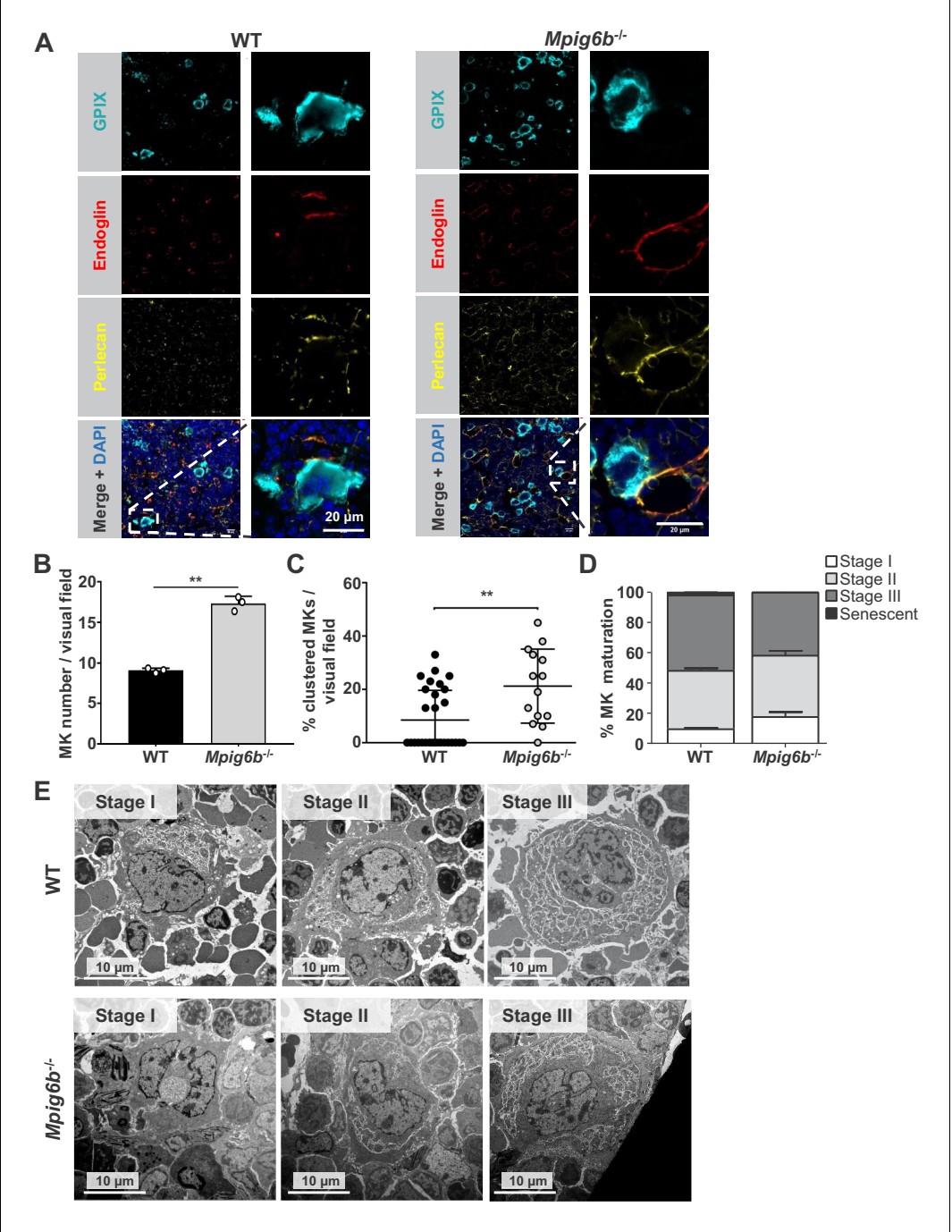

**Figure 9.** Megakaryocytes come into contact with perlecan in the bone marrow. (**A**) Analysis of immunofluorescent images of murine femur sections from WT and *Mpig6b*−/−mice. Sinusoids were marked using anti-endoglin (CD105) and MKs by anti-GPIX antibodies. Perlecan is abundantly expressed within the bone marrow cavity, present in intersinusoidal spaces and part of basement membranes in sinusoids and arterioles. MKs come into contact with perlecan. Scale bar: 20 µm. (**B**) Quantification of MKs in the bone marrow of WT and *Mpig6b*−/−mice; three animals of each genotype with five images per animal were analyzed. (**C**) Analysis of MK clustering, with % of clustered MKs per visual field with a total number of three mice per genotype analyzed; P values were calculated with Mann-Whitney U-test **, p<0.01. (**D**) Classification of the MK according to their maturation stage: stage I (absence of granules), stage II (granules and developing demarcation membrane system (DMS) not yet organized), stage III (DMS organized in cytoplasmic territories). Data are reported as the percentage of the total number of MK. Bars represent the mean ± SEM in three bone marrow samples (total number of MK

*Figure 9 continued on next page*

*Figure 9 continued*

counted 395–469). (E) Representative transmission electron microscopy (TEM) images of bone marrow from WT and *Mpig6b*$^{-/-}$ mice. Bars: 10 μm.

DOI: https://doi.org/10.7554/eLife.46840.030

The following figure supplements are available for figure 9:

**Figure supplement 1.** Overview sections of the bone marrow from WT and *Mpig6b*$^{-/-}$mice.
DOI: https://doi.org/10.7554/eLife.46840.031
**Figure supplement 2.** *Mpig6b*$^{-/-}$megakaryocytes form clusters.
DOI: https://doi.org/10.7554/eLife.46840.032

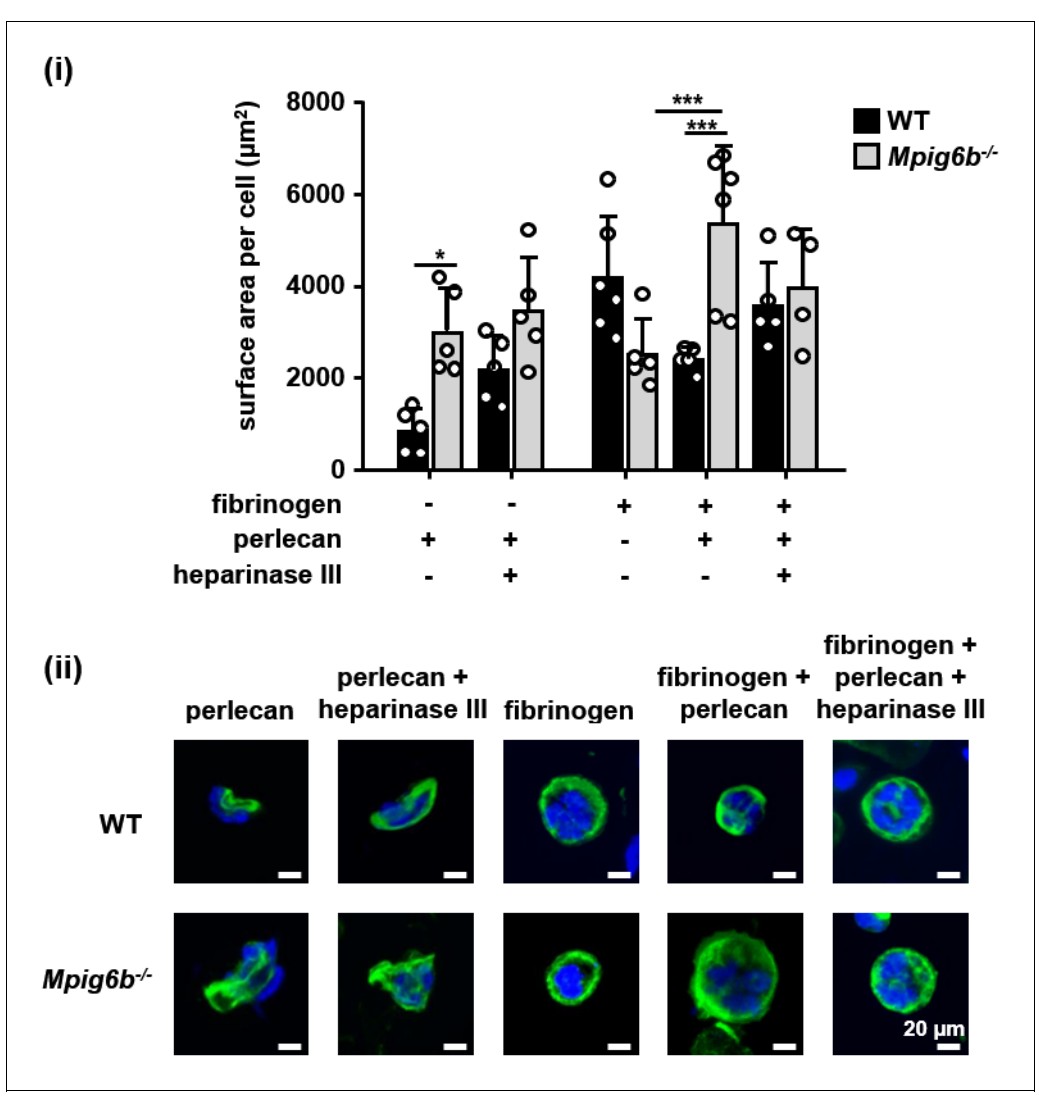

**Figure 10.** G6b knockout megakaryocytes show enhanced spreading on perlecan. Adhesion of WT and *Mpig6b*$^{-/-}$MKs on perlecan. (i) Mean surface area of MKs was quantified with ImageJ. n = 4–6 mice/condition/genotype from three independent experiments; total cell numbers analyzed per condition/genotype were 77–188 for conditions with perlecan only and 1671–2866 for conditions with fibrinogen. P values were calculated using two-way ANOVA with Sidak's post-hoc test, ***, p<0.001; *, p<0.05. (ii) Representative images of platelets stained for tubulin (green) and DAPI (blue); scale bar: 20 μm.

DOI: https://doi.org/10.7554/eLife.46840.033

To corroborate that the APAC-induced increase in Shp1 and Shp2 phosphorylation are mediated by G6b-B, we conducted signaling experiments in platelets from WT and *Mpig6b*$^{-/-}$ mice. APAC treatment of WT platelets recapitulated the effects observed in human platelets, showing only a modest change in overall phosphorylation pattern, and an increase in Shp1 and Shp2 phosphorylation (*Figure 13C,D*). By contrast, APAC-induced robust tyrosine phosphorylation in G6b-B-deficient platelets (*Figure 13C*), indicative of reduced inhibitory signaling and platelet hyperreactivity the absence of G6b-B. Strikingly, this was accompanied by reduced tyrosine phosphorylation of Shp1 and Shp2 in these platelets compared with WT platelets (*Figure 13D*). Collectively, these findings demonstrate that heparin and APAC have a direct effect on G6b-B phosphorylation, however, only the high-density ligand APAC is able to induce robust downstream inhibitory signaling via G6b-B, culminating in Shp1 and Shp2 binding and tyrosine phosphorylation.

## Discussion

In this study, we present evidence that establishes G6b-B as a functional receptor of HS and heparin. Little was known about the effects of GAGs on platelet and megakaryocyte function and the underlying molecular mechanisms, thus these findings represent a major advance in our understanding of the interaction, and of the biological and biochemical effects, of GAGs on these cells. Using a mass-spectrometry-based approach and subsequent in vitro binding assays, we identified the HS chains of perlecan as a physiological binding partner of G6b-B. The binding of G6b-B to HS was corroborated by a cell-based CRISPR KO screening, which identified molecules involved in the HS synthesis pathway as a prerequisite of G6b-B binding. There are two possible explanations as to why this assay did not identify perlecan, nor any other individual HSPGs as binding partners of G6b-B: first, the CRISPR screening approach will not identify genes that are essential for cell viability; and second, it will not identify proteins that have redundant functions. Given that perlecan is secreted from endothelial and smooth muscle cells, it is possible that there could be HSPGs other than perlecan (syndecans/glypicans) on the cell surface that carry the GAG chains in HEK cells. As the molecules in the HS synthesis pathway are essential for their respective synthesis, they can be identified in this approach more easily. This potential redundancy of HSPGs may also exist in vivo, and we cannot exclude the possibility that G6b-B may interact with other HSPGs in the cardiovascular system.

As with many other HS-binding molecules, G6b-B also binds structurally related heparin (*Xu and Esko, 2014*). Indeed, the interaction between heparin and G6b-B had been described previously, but the molecular details of the interaction and their functional significance had not been determined (*de Vet et al., 2005*). HS chains are not homogenously sulfated; instead, highly sulfated residues are clustered in domains along the polymer (called N-sulfated (NS) or sulfated (S) domains), which are interspersed by stretches of N-acetylated disaccharides (NA domains) that are largely devoid of sulfate groups (*Murphy et al., 2004*; *Xu and Esko, 2014*). Heparin, a degradation product derived from HS isolated from porcine intestine, shows larger NS domains and a greater degree of sulfation than HS. Hence, it is often used as an analogue for the NS domains of HS, despite the limitation that it may lack the protein binding properties of less sulfated HS. Our structural analysis of the G6b-B ligand complex shows that G6b-B interacts with multiple sulfates in the heparin oligosaccharide. Hence several observations, including the higher potency of heparin in inducing G6b-B phosphorylation as compared to HS, and the shift in dose-response curve in aggregometry, may be due to the larger NS domains in the heparin molecule. In addition, synthesis of HS is not template-driven, therefore, the length and distribution of such domains is regulated in a tissue- or cell-specific manner, adding additional complexity to the regulatory role of HS.

Our size-exclusion chromatography data demonstrate that the dimerization of G6b-B is induced by the heparin ligand. The crystal structure of heparin-bound G6b-B reveals the mode of ligand binding and how the binding of this ligand induces ectodomain dimerization. The contact surfaces between the G6b-B dimer and the Fab fragments are spatially separated from the heparin-binding site, suggesting that the presence of the Fab fragments does not interfere with heparin binding. Heparin-dependent, non-constitutive dimerization of G6b-B is consistent with the small interface between the G6b-B subunits and the absence of main chain-main chain hydrogen bonds across the β-sheet of the binding surface. Among 34 entries currently in the PDB of structures containing heparin as a ligand, dimeric assemblies (or multimeric assemblies with a two-fold rotation axis) are common (*Figure 14—figure supplement 1*), but the anti-parallel alignment of two Ig-like domains in the

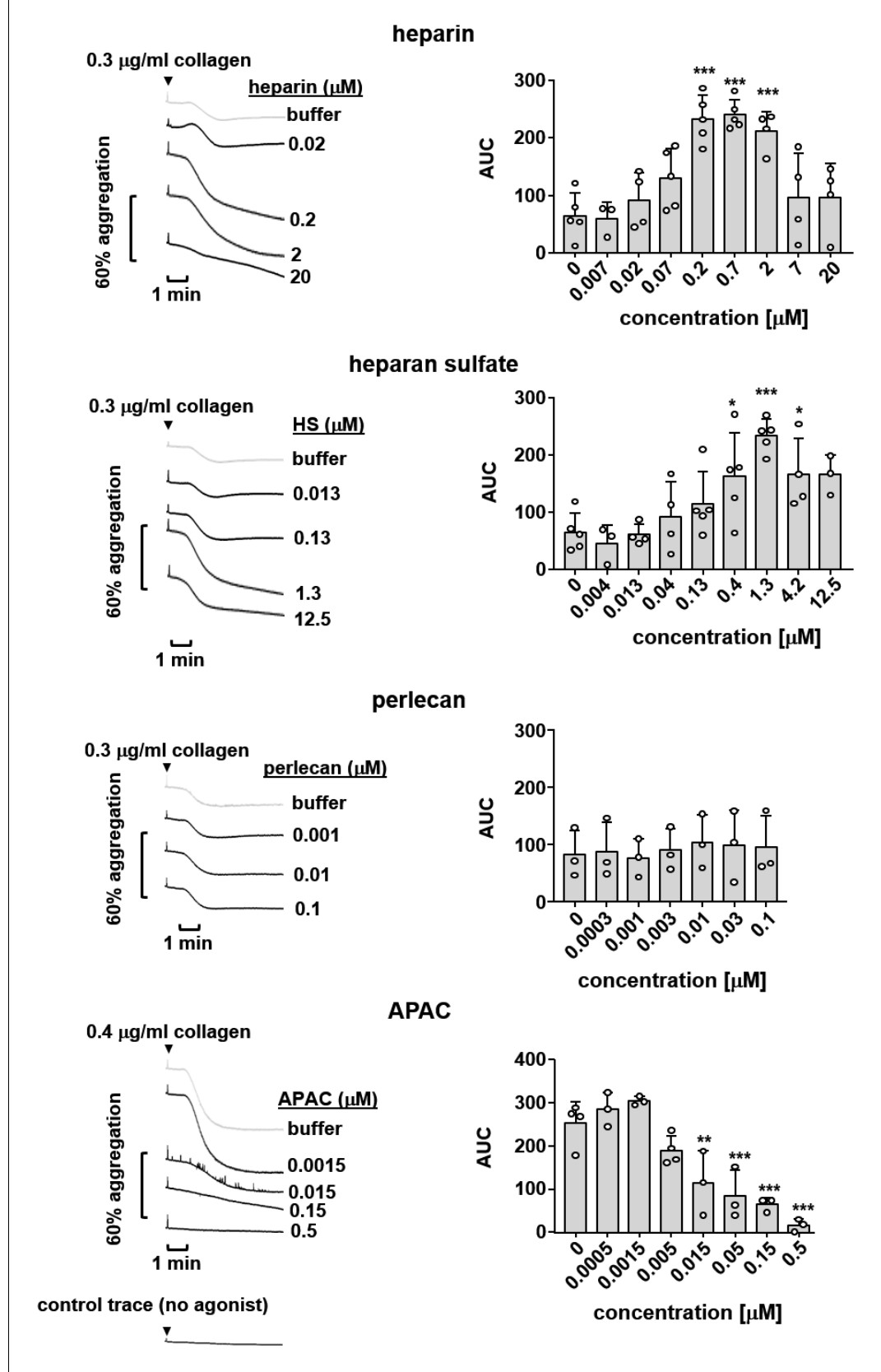

**Figure 11.** Effects of G6b-B ligands on platelet aggregation. Human platelet rich plasma (PRP) was incubated with the indicated compound for 90 s prior to agonist addition. Aggregation traces were recorded on a Chronolog four

*Figure 11 continued on next page*

*Figure 11 continued*

channel aggregometer. Averaged aggregation traces (left) and area under the curve (AUC) quantification (right) of platelet aggregation (n = 3–5 per condition). P-values were calculated using one-way ANOVA with Dunnett's post-hoc test and refer to the untreated control. ***, p<0.001; **, p<0.01; and *, p<0.05.

DOI: https://doi.org/10.7554/eLife.46840.034

---

heparin-bound structure of G6b-B appears to be unique (*Cai et al., 2015*; *Dahms et al., 2015*; *Fukuhara et al., 2008*; *Pellegrini et al., 2000*; *Schlessinger et al., 2000*). The involvement of the β-sheet surface in heparin binding is somewhat reminiscent of how carbohydrate-binding modules (CBM) bind saccharide ligands (*Abbott and van Bueren, 2014*). CBMs are non-enzymatic domains often associated with carbohydrate-active enzymes, which contribute to carbohydrate binding and discrimination (*Boraston et al., 2004*).

The crystal structure of G6b-B shows a prominent positively charged electrostatic surface area, but this positive surface patch runs perpendicular to the central cleft of the G6b-B dimer. Indeed, the heparin oligosaccharide lines up with the cleft, rather than extending along the positive surface patch. Comparison with other heparin-bound structures (*Figure 14—figure supplement 1*) suggests that charge complementation is not the sole determinant of the mode of heparin binding, and that the depth and shape of the docking site are likely to be important as well. Nevertheless, charge complementing ionic interactions lock the ligand in to register at the center of the G6b-B binding cleft, where the sparsity of sulfate-Arg or sulfate-Lys interactions is surprising. The crystal structure rationalizes the diminished binding of G6b-B transfected HEK293 cells to biotinylated heparin when the four basic residues Lys54, Lys58, Arg60 and Arg61 are simultaneously mutated. Among these four side-chains, the key interaction appears to be with Arg60, as Arg61 is shielded through G6b-B dimerization from the ligand, Lys54 is well separated from the binding cleft and Lys58 is situated within a 4 Å-radius of heparin, but makes no polar interactions. The heparin ligand does not exhaust the possibilities for specificity-determining interactions with G6b-B in the ligand-binding cleft. For instance, Arg60[F] and Lys109[F], but not their counterparts in chain E on the opposite side of the cleft, are involved in ionic interactions with the same sulfate group. It is conceivable that the physiological HS ligand of G6b-B may have a different pattern of sulfate groups that engage both Arg60 and Lys109, perhaps in addition to Lys58.

Since G6b-B shows a considerable degree of glycosylation, the question arises as to whether this might modulate the ligand interaction. Through the course of our structural analysis of G6b-B, we identified multiple glycosylation sites (N32, S67, S68, S69, T71, T73) in the G6b-B ectodomain. Although most of these sites can be mutated to increase protein homogeneity for crystallization studies, the final structure of the G6b-B–heparin complex revealed that all of these glycosylation sites are spatially separated from the ligand-binding surface, and are not likely to impede ligand binding sterically. All of the recombinant G6b-B molecules used in this study were produced in mammalian cell lines and are therefore glycosylated. We previously showed that MK and platelet G6b-B migrate at the same molecular weight by Western blotting (*Mazharian et al., 2012*), suggesting that G6b-B is not differentially glycosylated in MKs and platelets. We currently have no evidence that the glycosylation of G6b-B alters ligand binding.

Investigating the functional consequences of this interaction revealed that heparin and HS have complex effects on platelet function and that G6b-B is a key regulator in this process. Our data demonstrates that, to induce robust inhibitory biological or signaling effects, G6b-B ligands need to be either immobilized to a surface, as in the case of perlecan-coated plates, or multivalent, as in the case of APAC. By contrast, single-chain heparin and HS enhanced rather than inhibited platelet aggregation. These findings are in line with numerous previous reports, showing enhancing effects of heparin on platelet aggregation in platelet-rich plasma (*Gao et al., 2011*; *Saba et al., 1984*; *Salzman et al., 1980*). This most likely also contributes to a mild drop in platelet counts in patients receiving heparin, referred to as non-immune heparin-induced thrombocytopenia (*Cooney, 2006*). On the basis of our signaling data and size-exclusion chromatography data, we assume that heparin, despite being able to dimerize the receptor, fails to cluster G6b-B sufficiently into higher-order oligomers to induce robust downstream signaling (*Figure 14A,B*). It remains to be determined whether the enhancing effects of heparin and HS on platelet aggregation is mediated by

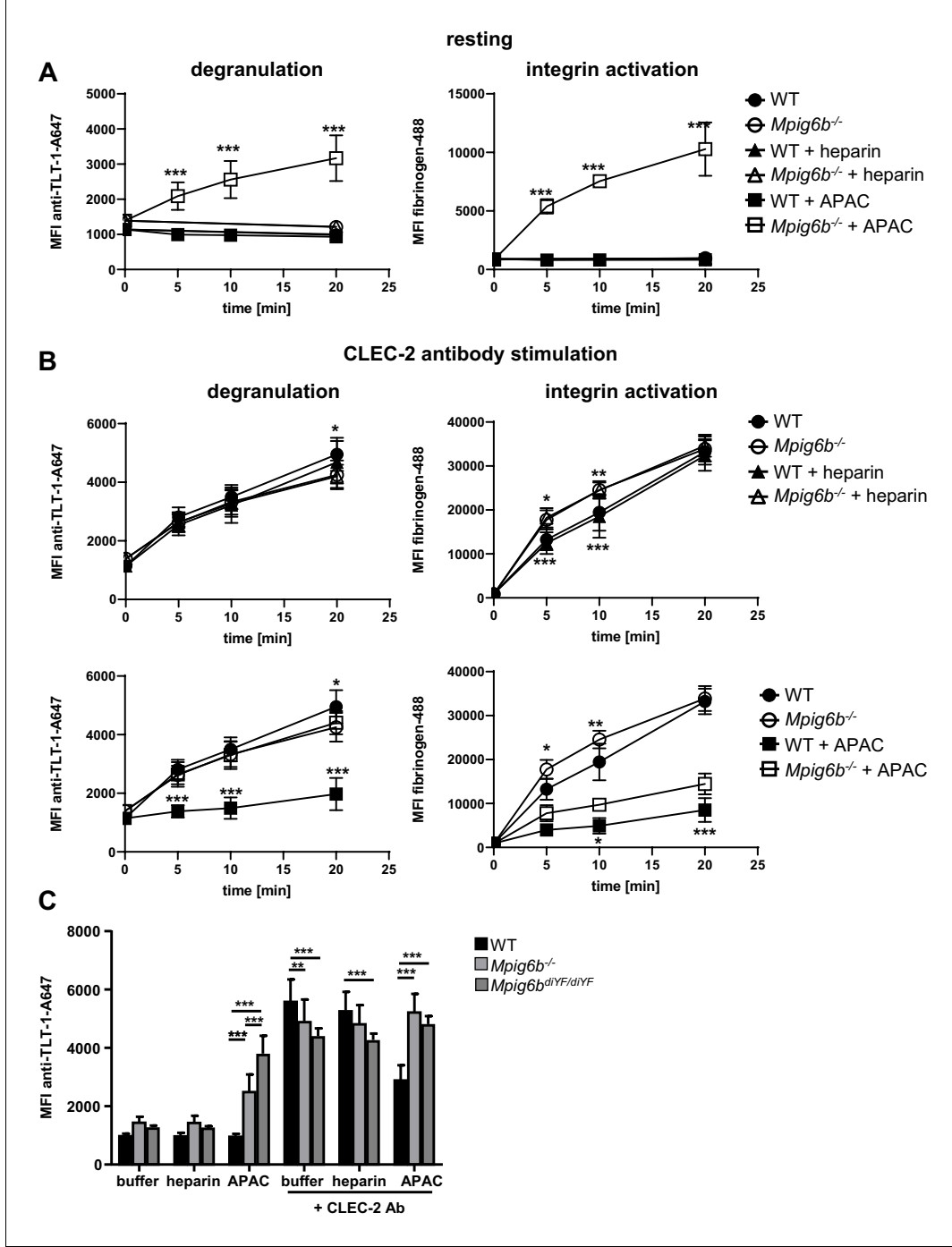

**Figure 12.** APAC inhibits CLEC-2-mediated degranulation in WT but not *Mpig6b* KO platelets. Mouse blood, diluted 1:10 in Tyrode's buffer was incubated with the indicated compounds (0.05 µM) in the (**A**) absence or (**B**) presence of a stimulating CLEC-2 (3 µg/ml) for the indicated time. Samples were fixed and TLT-1 surface levels, a marker for platelet degranulation or fibrinogen-Alexa488 binding (a measure of integrin activation), were determined by flow cytometry. n = 5–6 mice/condition/genotype from two independent experiments. P-values were calculated using (**A**) two-way ANOVA with Sidak's post-hoc test (comparison of WT APAC vs *Mpig6b*$^{-/-}$ APAC) or (**B**) two-way ANOVA with Tukey's post-hoc test, and refer to the difference between WT and *Mpig6b*$^{-/-}$. ***, $p<0.001$; **, $p<0.01$; and *, $p<0.05$. (**C**) Mouse blood was incubated with the indicated compounds for 20 min. Samples were fixed and TLT-1 surface levels were determined by flow cytometry. n = 6–8 mice/condition/genotype from three independent experiments. P-values were calculated using two-way ANOVA with Tukey's post-hoc test. ***, $p<0.001$; and **$p<0.01$. Source data are available in *Figure 12—source data 1*.
*Figure 12 continued on next page*

*Figure 12 continued*

DOI: https://doi.org/10.7554/eLife.46840.035

The following source data is available for figure 12:

**Source data 1.** Source data for graphs shown in *Figure 12A–C*.

DOI: https://doi.org/10.7554/eLife.46840.036

reducing the inhibitory effects of G6b-B alone or by additional effects on other platelet receptors, which promote platelet activation, such as the integrin αIIbβ3, previously shown to bind heparin (*Fitzgerald et al., 1985*; *Gao et al., 2011*; *Sobel et al., 2001*).

In contrast to these soluble, monovalent ligands, the HS side-chains of immobilized perlecan exerted an inhibitory effect on platelets, as evidenced by the impaired adhesion of platelets to collagen and fibrinogen. This extends observations from previous reports describing the anti-adhesive properties of the HS chains of perlecan, although the underlying mechanism was not known (*Klein et al., 1995*; *Lord et al., 2009*). Moreover, heparinized polymers showed less platelet adhesion than their non-heparinized counterparts (*Han et al., 1989*; *Lindhout et al., 1995*; *Olsson et al., 1977*). Our results with platelets from G6b-B-deficient mice demonstrate that heparin or HS engagement by G6b-B on these surfaces induces an inhibitory signal, blocking platelet activation and adhesion.

The failure of perlecan to have an effect on collagen-mediated platelet aggregation and platelet signaling in solution might simply be a consequence of the test conditions and suggests that it must be immobilized to a surface in order to present HS chains at a density sufficient to induce the inhibitory effects observed. Hence, to determine the effect of G6b-B clustering in solution, we took advantage of APAC, which mimics naturally occurring macromolecular heparin proteoglycans and harbors a higher GAG density than perlecan. On the basis of our findings, and given the dependence of G6b-B binding to sulfate groups and the common use of heparin as an analog for the NS domains of HS (*Xu and Esko, 2014*), we concluded that the inhibitory effect of both APAC and the HS chains of perlecan is mediated by clustering G6b-B. We compared the effect of APAC to that of single-chain heparin, which differ only by the clustering of the chains, but not the structure or composition of the polysaccharide chains. Similar to previous reports (*Kauhanen et al., 2000*; *Lassila and Jouppila, 2014*; *Lassila et al., 1997*), we found that APAC inhibited platelet activation via the ITAM-containing GPVI-FcR γ-chain receptor complex, and also via the hemi-ITAM-containing receptor CLEC-2. Thus, by increasing the clustering capacity of heparin to a multivalent form, an inhibitory effect on platelet function was achieved in solution. In line with this observation, we found that APAC induced stronger phosphorylation of G6b-B, which was accompanied by the association and phosphorylation of the tyrosine phosphatases Shp1 and Shp2, which is not observed in G6b-B-deficient platelets. We therefore conclude that clustering of G6b-B receptor dimers into higher-order oligomers by an immobilized or multivalent ligand is required for the inhibitory effect on platelet function (*Figure 14C*).

Perlecan is secreted by endothelial and smooth muscle cells into the extracellular space of the vessel wall and hence is inaccessible by platelet G6b-B in an intact blood vessel (*Murdoch et al., 1994*; *Saku and Furthmayr, 1989*; *Segev et al., 2004*). Only upon vascular injury will the interaction between platelet G6b-B and perlecan occur, resembling the interaction of other platelet receptors with ligands that are expressed in the vessel wall, including collagen, laminin and fibronectin (*Bergmeier and Hynes, 2012*). Given the results of our adhesion assay, we speculate that the interaction of platelet G6b-B with perlecan negatively regulates the initial steps of thrombus formation, preventing thrombi from forming unnecessarily.

The G6b-B-HS interaction may also be relevant for triggering the directional formation of proplatelets by MKs towards sinusoidal blood vessels at sites of platelet production. A key, yet unresolved, question is how MKs remain relatively refractory and do not release platelets into the ECM-rich environment of the bone marrow despite expressing the same repertoire of cell-surface receptors as platelets. G6b-B is highly expressed in mature MKs and G6b-B KO and loss-of-function mice show a severe macrothrombocytopenia resulting from impaired proplatelet formation and platelet production, which is accompanied by an increase in MK numbers in the bone marrow (*Geer et al., 2018*; *Mazharian et al., 2012*). Here, we provide first evidence for a potential role of perlecan in regulating

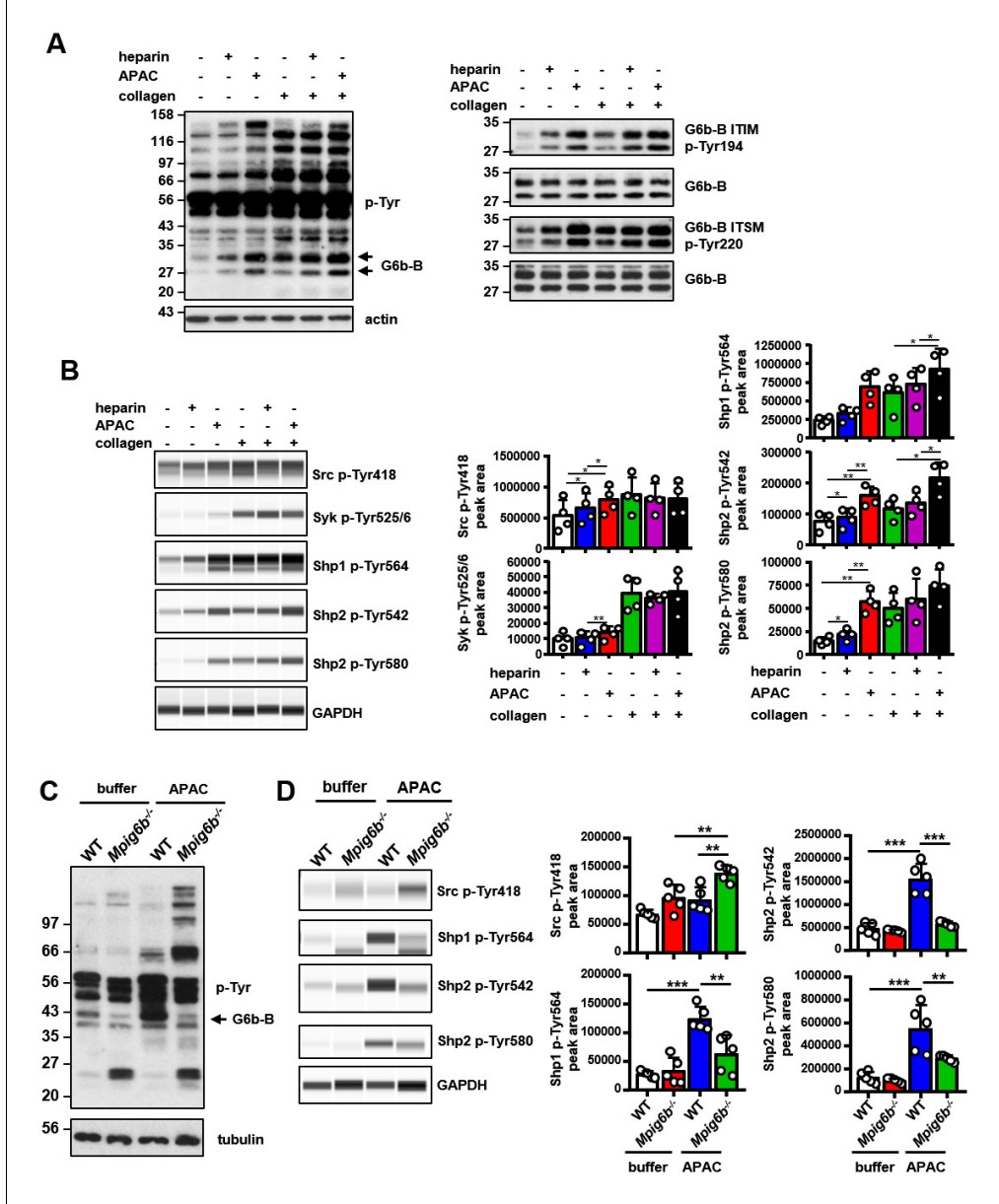

**Figure 13.** APAC induces G6b-B phosphorylation and downstream signaling. (**A**) Washed human platelets (5 × $10^8$/ml) were incubated for 90 s with 0.05 µM APAC, 0.7 µM heparin or buffer in the presence of 10 µM integrilin. Where indicated, platelets were additionally stimulated with 3 µg/ml collagen for 90 s following compound treatment. Samples were lysed and whole cell lysates (WCL) were analyzed by western blotting. Representative western blots are from n = 3–5 independent experiments. (**B**) Lysates were also analyzed by quantitative capillary-based gel electrophoresis with the indicated antibodies. Representative data are displayed as blots on the left and as quantified peak areas on the right. (**C, D**) Washed mouse platelets (5 × $10^8$/ml) were incubated for 90 s with 0.05 µM APAC or buffer in the presence of 10 µM lotrafiban. Samples were analyzed as described above. The *Mpig6b*$^{-/-}$ samples show IgG light chain fragments, which result from IgG binding to the platelet surface. P-values were calculated using one-way ANOVA with Sidak's post-hoc test. ***, p<0.001; **, p<0.01; and *, p<0.05. p-Tyr, anti-phosphotyrosine (4G10).

DOI: https://doi.org/10.7554/eLife.46840.037

The following figure supplement is available for figure 13:

**Figure supplement 1.** Effects of G6b-B ligands on G6b-B phosphorylation.

DOI: https://doi.org/10.7554/eLife.46840.038

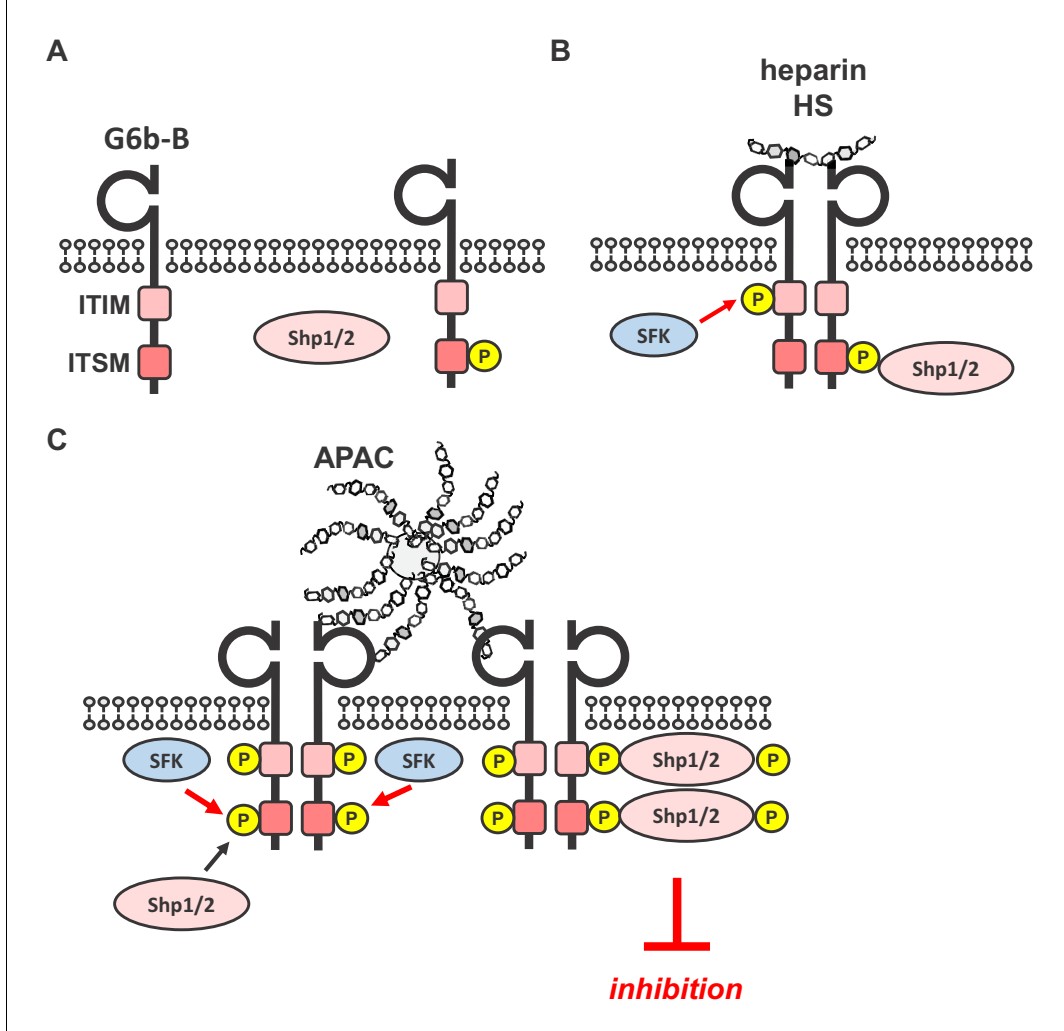

**Figure 14.** Simplified model of glycan-mediated regulation of G6b-B function. (**A**) In the absence of any ligand, G6b-B is mainly present in a monomeric state and phosphorylated to a low degree. (**B**) Small soluble ligands, for example heparin, induce the dimerization of the receptor, but induce only mild G6b-B phosphorylation and downstream signaling. (**C**) Multivalent ligands, for example the HS chains of vessel-wall perlecan (not shown) or the heparin molecules in APAC, cluster G6b-B dimers into higher-order oligomers. Hence, they facilitate downstream signaling of G6b-B, including robust phosphorylation of G6b-B and downstream Shp1 and Shp2 phosphatases, resulting in the inhibition of platelet activation. SFK, src family kinase.

DOI: https://doi.org/10.7554/eLife.46840.039

The following figure supplement is available for figure 14:

**Figure supplement 1.** Selected structures of proteins with a heparin ligand.

DOI: https://doi.org/10.7554/eLife.46840.040

MK function by demonstrating that G6b-B-deficient, but not WT, MKs increase their size in the presence of the HS side-chains of perlecan, indicating cellular activation. Because it has been shown in vitro (**Mazharian et al., 2012**) and ex vivo (this publication) that G6b-B-deficient MKs have no overt maturation defect and as an increase in size was only observed in the presence of the HS chains of perlecan, it seems unlikely that the increase in size results from an overall maturation defect of the MKs. In addition, findings from our study and others demonstrate that perlecan is abundantly expressed in the bone marrow ECM and comes into contact with mature MKs (**Farach-Carson et al., 2014**). We hypothesize that the MK G6b-B-HS interaction might play a role in regulating polarized proplatelet formation in the sinusoidal blood vessel lumen.

Despite sharing a common structure, the biophysical details of HS are tissue- or cell-specific, including variability of sulfation patterns, suggesting that the regulatory role of HS occurs in a spatio-temporal manner in different tissues and at different developmental and pathological stages. Hence, future research is needed to investigate whether and to what extent G6b-B interacts with other HS proteoglycans and to determine the physiological relevance of these interactions under normal and pathological conditions, such as cardiovascular disease in which the composition of the vascular glycocalyx is altered (*Kim et al., 2017*). Moreover, our results also demonstrate that platelets and MKs from G6b-B-deficient mice showed an activation response towards the HS chains of immobilized perlecan and, in case of platelets, also towards APAC, even in the absence of a classical platelet agonist such as collagen. Hence, one of the key functions of G6b-B in vivo may not be solely restricted to the inhibition of platelet function upon vascular injury, but may also retain platelets in a resting state by inhibiting other HS-binding platelet receptors, which remain to be identified. Notably, the HSPGs syndecan-1 and −4 are expressed on the surface of endothelial cells that form an integral part of the glycocalyx (*Marki et al., 2015*). As platelets marginate to the vessel wall, the interaction of G6b-B on circulating platelets within the glycocalyx may induce a low level inhibitory signal that helps to maintain platelets in an inactive state, in line with the basal phosphorylation of G6b-B in resting platelets.

In summary, our findings establish the interaction of G6b-B with heparan sulfate as a novel mechanism regulating platelet reactivity, and demonstrate important implications of this interaction in the regulation of platelet production and the adverse effects upon soluble heparin administration.

# Materials and methods

## Key resources table

| Reagent type (species) or resource | Designation | Source or reference | Identifiers | Additional information |
|---|---|---|---|---|
| Genetic reagent (*Mus musculus*) | *Mpig6b*⁻/⁻ | PMID: 23112346 | | Dr. Yotis Senis (University of Birmingham and EFS Grand Est, Inserm UMR-S1255) |
| Genetic reagent (*M. musculus*) | *Mpig6b*^{diYF/diYF} | PMID: 29891536 | | Dr. Yotis Senis (University of Birmingham and EFS Grand Est, Inserm UMR-S1255) |
| Cell line (*Cricetulus griseus*) | A5 CHO | other | | provided by Dr. Ana Kasirer-Friede and Dr. Sanford Shattil (University of California, San Diego) |
| Antibody | anti-perlecan (rat monoclonal) | Santa Cruz Biotechnologies | clone A7L6; sc-33707; RRID:AB_627714 | (1:100); used for IF staining of bone marrow (BM) |
| Antibody | anti-mouse CD105 (Endoglin) (rat monoclonal) | eBioscience/Thermo Fisher Scientific | #MA5-17943; clone MJ7/18; RRID:AB_2539327 | (1:100); used for IF staining of BM |
| Antibody | anti-GPIX-Alexa488 (rat monoclonal) | other | clone 56F8 | 1.4 µg/ml; used for IF staining of BM, custom made lab reagent |
| Antibody | anti rat IgG Alexa 647 (goat polyclonal) | Invitrogen | #A-21247; RRID:AB_141778 | (1:300); used for IF staining of BM |

*Continued on next page*

*Continued*

| Reagent type (species) or resource | Designation | Source or reference | Identifiers | Additional information |
|---|---|---|---|---|
| Antibody | anti rat IgG Alexa 546 (goat polyclonal) | Invitrogen | #A-11081; RRID:AB_141738 | (1:300); used for IF staining of BM |
| Antibody | anti-actin (mouse monoclonal) | Sigma-Aldrich | #A4700, clone AC-40; RRID:AB_476730 | (1:1000) |
| Antibody | Anti-α-tubulin (mouse monoclonal) | Sigma-Aldrich | #T6199, clone DM1A; RRID:AB_477583 | (1:1000) |
| Antibody | anti-GAPDH (rabbit monoclonal) | Cell Signaling Technology | #2118, clone: 14C10; RRID:AB_561053 | (1:10) dilution, on 0.05 mg/ml lysates for Wes |
| Antibody | anti-Src p-Tyr418 (rabbit polyclonal) | Sigma-Aldrich | #44660G; RRID:AB_1500523 | (1:10) dilution, on 0.05 mg/ml lysates for Wes |
| Antibody | anti-Shp1 p-Tyr564 (rabbit monoclonal) | Cell Signaling Technology | #8849, clone: D11G5; RRID:AB_11141050 | (1:10) dilution, on 0.2 mg/ml lysates for Wes |
| Antibody | anti-Shp2 p-Tyr542 (rabbit polyclonal) | Cell Signaling Technology | #3751; RRID:AB_330825 | (1:10) dilution, on 0.2 mg/ml lysates for Wes |
| Antibody | anti-Shp2 p-Tyr580 (rabbit polyclonal) | Cell Signaling Technology | #3703; RRID:AB_2174962 | (1:10) dilution, on 0.2 mg/ml lysates for Wes |
| Antibody | anti-Syk p-Tyr525/6 (rabbit polyclonal) | Cell Signaling Technology | #2711; RRID:AB_2197215 | (1:50) dilution, on 0.2 mg/ml lysates for Wes |
| Antibody | anti-SH-PTP1/Shp-1 (rabbit polyclonal) | Santa Cruz | sc-287 (C19); RRID:AB_2173829 | (1:1000) |
| Antibody | anti-SH-PTP2/Shp-2 (rabbit polyclonal) | Santa Cruz | sc-280 (C18); RRID:AB_632401 | (1:1000) |
| Antibody | anti-phosphotyrosine (mouse monoclonal) | Merck-Millipore | 05–321, clone 4G10; RRID:AB_309678 | (1:1000) |
| Antibody | anti-human G6b-B (mouse monoclonal) | other | clone 17–4 | 10 µg/ml, custom-made lab reagent |
| Peptide, recombinant protein | purified human IgG-Fc fragment | Bethyl Laboratories | P80-104 | |
| Peptide, recombinant protein | recombinant Mouse Syndecan-2/CD362 protein, CF | R&D Systems | 6585-SD-050 | |
| Peptide, recombinant protein | recombinant human Agrin protein, N-terminal, CF | R&D Systems | 8909-AG-050 | |
| Peptide, recombinant protein | rec. human laminin 111 | Biolamina | LN111-02 | |
| Peptide, recombinant protein | rec. human laminin 411 | Biolamina | LN411-02 | |
| Peptide, recombinant protein | rec. human laminin 421 | Biolamina | LN421-02 | |
| Peptide, recombinant protein | rec. human laminin 511 | Biolamina | LN511-02 | |

*Continued on next page*

Continued

| Reagent type (species) or resource | Designation | Source or reference | Identifiers | Additional information |
|---|---|---|---|---|
| Peptide, recombinant protein | rec. human laminin 521 | Biolamina | LN521-02 | |
| Chemical compound, drug | heparan sulfate proteoglycan | Sigma-Aldrich | H4777 | alternative name: perlecan |
| Chemical compound, drug | heparin | Iduron | HEP001 | https://iduron.co.uk/product/Heparin-1 |
| Chemical compound, drug | heparin oligosaccharide dp4 | Iduron | HO04 | https://iduron.co.uk/product/Heparin-1 |
| Chemical compound, drug | heparin oligosaccharide dp8 | Iduron | HO08 | https://iduron.co.uk/product/Heparin-1 |
| Chemical compound, drug | heparin oligosaccharide dp12 | Iduron | HO12 | https://iduron.co.uk/product/Heparin-1 |
| Chemical compound, drug | heparin oligosaccharide dp20 | Iduron | HO20 | https://iduron.co.uk/product/Heparin-1 |
| Chemical compound, drug | 2-O-desulphated heparin | Iduron | DSH001/2 | |
| Chemical compound, drug | 6-O-desulphated heparin | Iduron | DSH002/6 | |
| Chemical compound, drug | N desulphated heparin | Iduron | DSH003/N | |
| Chemical compound, drug | N-desulphated re N-acetylated heparin | Iduron | DSH004/Nac | |
| Chemical compound, drug | heparan sulphate | Iduron | GAG-HS01 | |
| Chemical compound, drug | HS fraction III approx. mol. wt. 9 kDa | Iduron | GAG-HS III | |
| Chemical compound, drug | APAC | Aplagon Oy | | |
| Chemical compound, drug | heparinase III (heparitinase I) *Flavobacterium heparinum* (EC 4.2.2.8) | AMSBiotechnology | AMS.HEP-ENZ III | |
| Chemical compound, drug | Heparin—biotin sodium salt | Sigma-Aldrich | B9806-10MG | |
| Chemical compound, drug | fibronectin | Cabiochem | Cat #341631 | |
| Chemical compound, drug | fibrinogen | Enzyme Research Laboratories | Fib 3 3496L | |
| Chemical compound, drug | collagen I | Takeda | 1130630 | collagen reagens horms |
| Chemical compound, drug | Cultrex Mouse Collagen IV | Trevigen | 3410-010-01 | purchased via R & D Systems |

*Continued*

| Reagent type (species) or resource | Designation | Source or reference | Identifiers | Additional information |
|---|---|---|---|---|
| Chemical compound, drug | Laminin from EHS murine sarcoma basement membrane | Sigma-Aldrich | L2020 | refers to mouse laminin-111 in this study |
| Chemical compound, drug | streptavidin | Sigma-Aldrich | S4762 | |
| Software | Cell Profiler (2.2.0) | Broad Institute | http://cellprofiler.org/ RRID:SCR_007358 | |
| Software | Fiji | PMID: 22743772 | https://imagej.net/Fiji; RRID:SCR_002285 | |

## Mice

*Mpig6b* (*Mpig6b$^{-/-}$*) and *Mpig6b$^{diY/F}$* knock-in (*Mpig6b$^{diYF/diYF}$*) mice were generated on a C57BL/6 background by Taconic Artemis (Cologne, Germany) as previously described (*Geer et al., 2018*; *Mazharian et al., 2012*). Control mice were pure C57BL/6 (*Mpig6b$^{+/+}$*), referred to as WT. All procedures were undertaken with UK Home Office approval (project license No P46252127) in accordance with the Animals (Scientific Procedures) Act of 1986.

## Reagents and antibodies

See Key Resources Table for information on the sources of key reagents used in this study.

p-nitrophenyl phosphate (pNPP) and goat anti-human IgG–HRP antibody were obtained from Sigma-Aldrich, Dorset, UK. The semisynthetic macromolecular conjugate of unfractionated heparin and a human serum albumin, APAC, was from Aplagon Oy, Helsinki, Finland. Blocking medium (2.5% horse serum) and 3,3'-diaminobenzidine tetrahydrochloride (DAB) peroxidase substrate for immunohistochemistry were purchased from Vector Laboratories, Peterborough, UK and 3,3,5,5 tetramethylbenzidine (TMB) was from BD Biosciences, Wokingham, UK. Polyclonal phospho-specific G6b-B antibodies were generated by Biogenes, Berlin, Germany. Phalloidin-Alexa 488 was from Invitrogen Life Technologies, Paisley, UK. All other antibodies and chemicals were either purchased or generated as previously described (*Mazharian et al., 2012*).

## Constructs

*Recombinant proteins:* the cDNA encoding the mouse G6b-B extracellular domain was amplified by PCR using the primers GATC AAGCTT ATG GCC TTG GTC CTG CCG CTG (forward) and GATC GGATCC ACT TAC CTG T CTC GTA CCC GTG GGT AGA TCC (reverse) from a mouse megakaryocyte cDNA library template. The PCR product was cleaved using Hind III and Bam HI and ligated into pCDNA3Ig, which was comprised of the genomic human IgG1 hinge-C2-C3 Fc region cloned into the HindIII and Not I sites of pcDNA3. This creates a construct encoding the extracellular part of G6b, spliced in frame with the IgG1 hinge, producing a G6b-B-Fc chimeric dimer. The resulting protein, mG6b-B-Fc, was expressed in COS-7 cells and then purified via affinity chromatography. The human G6b-B-Fc dimer (hG6b-B-Fc) construct was produced using an identical approach to the murine construct, using the primers GATC AAGCTT ATG GCT GTG TTT CTG CAG CTG (forward) and GATC GGATCC ACTTACCTGT CTG GGG ATA CAC GGA CCC ATG (reverse). Similarly, untagged monomeric G6b-B (residues 18–142) as well as His-tagged versions were produced — human G6b-B (residues 18–142)-Fc-His6 (expressed as a homodimer) and human G6b-B (residues 18–142)-Fc-His6/Fc-streptagII (heterodimer, monomeric for G6b-B; Peak Proteins Limited, Alderley Park) — for use in surface plasmon resonance measurements. All human constructs were expressed transiently in HEK293-6E cells.

*Cell culture:* The cDNA encoding the full length of human G6b-B protein was amplified by PCR from a human cDNA library. This PCR fragment was first cloned into the pCR-Blunt vector (Invitrogen), and then subcloned into the pCDNA3 vector, for the expression of untagged G6b-B in heterologous cell systems. Subsequently, the G6b-B mutant that is mutated in the potential heparin

binding site (hG6b-B K54D/K58D/R60E/R61E) was generated with the Quick Change Site-directed mutagenesis kit (Agilent Technologies, Stockport, UK).

## Immunohistochemistry

Immunohistochemistry stainings were performed according to standard protocols. In brief, frozen mouse tissue sections (Zyagen, San Diego, CA, USA) were thawed and washed once in phosphate buffered saline (PBS). After blocking for 20 min (min) at room temperature (RT), tissues were incubated with mG6b-B-Fc or human IgG-Fc fragment (negative control, 5 µg/ml in PBS) for 75 min at RT. After three washing steps in PBS, slides were fixed in acetone/PFA for 4 min and endogenous peroxidase was blocked with 3% $H_2O_2$ in methanol (5 min). Slides were incubated with anti-human IgG–HRP antibody (1:600 in PBS, 0.1% Tween 20) and the signal developed with DAB substrate. Subsequently, tissue sections were counterstained with hematoxylin. Images were acquired on a Zeiss Axio Scan.Z1 (Zeiss, Cambridge, UK) equipped with an 3CCD color 2MP Hitachi 1200 $\times$ 1600 HV-F202SCL camera, using a 10x (NA 0.45) or 20x (NA 0.8) plan apochromat air objective. Images were acquired and exported with the Zeiss Zen software.

## Femur sectioning and staining

Femurs of mice aged 6–12 weeks were sectioned and stained as described previously (*Kawamoto, 2003*; *Semeniak et al., 2016*). In brief, femora were isolated, fixed for 4 hr at 4°C in 4% PFA before being transferred along a sucrose gradient from 10%, 20% and 30%, each for 24 hr. Next, femora were embedded in SCEM medium (Section lab, Hiroshima, Japan) and frozen at −80°C until sectioning. Megakaryocytes were stained with anti-GPIX-Alexa488 antibody, and CD105 was used as an endothelial cell marker. Additional stainings were performed using antibodies against perlecan. Corresponding secondary antibodies were used at a 1:300 dilution. Slides were mounted in Fluoromount G including DAPI (Thermo Fisher Scientific). Recording was performed using a Leica TCS SP8 confocal laser scanning microscope (Leica, Wetzlar, Germany) with an 40x oil objective at 20°C. The numerical aperture (NA) of the objective lens was 1.3 and the software used for data acquisition was LAS X (Leica). Subsequently, images were processed with ImageJ (NIH, Bethesda, MD, USA). No 3D reconstruction, gamma adjustments or deconvolution were performed.

For reconstruction of whole femora sections, single images were taken with a resonant scanner and digitally stitched using LAS X software.

## Electron microscopy studies

Bone marrow samples obtained by flushing mouse femora with 0.1 M sodium cacodylate buffer were fixed in 2.5% glutaraldehyde and embedded in Epon as described (*Eckly et al., 2012*). Thin sections were stained with uranyl acetate and lead citrate, and examined under a JEOL 2100Plus transmission electron microscope at 120 kV (Jeol, Tokyo, Japan). Megakaryocytes were counted on whole transversal sections and the number of cells was expressed as a density per unit area (defined as one square of the grid, i.e. 13 000 µm$^2$). Megakaryocytes at stages I, II and III were identified using distinct ultrastructural characteristics: stage I, a cell 10–50 µm in diameter with a large nucleus; stage II, a cell 20–80 µm in diameter containing platelet-specific granules; stage III, a megakaryocyte containing a well-developed demarcation membrane system defining cytoplasmic territories and a peripheral zone. Samples from three mice of each genotype were examined in each case.

## Pull-down and identification of the ligand

Venae cavae were dissected from wild-type mice and fat and connective tissue were removed. The endothelial tissue was placed in lysis buffer (10 mM Tris-HCl (pH 7.6), 150 mM NaCl, 1 mM EGTA, 1 mM EDTA, 1% IGEPAL CA-630, 5 mM $Na_3VO_4$, 0.5 mM 4-(2-aminoethyl) benzenesulfonyl fluoride hydrochloride, 5 µg/ml leupeptin, 5 µg/ml aprotinin, 0.5 µg/ml pepstatin) and homogenized with a PowerGen homogenizer (Fisher Scientific, Loughborough, UK). Lysates were centrifuged at 13,000 $\times$ g for 10 min at 4°C. Supernatants were collected and re-centrifuged under the same conditions. Protein lysate was precleared with Protein G sepharose (PGS, 50% slurry) and human IgG-Fc fragment by agitation for 1 hr at 4°C. The lysate was then split into two samples which received either mG6b-B-Fc or human IgG-Fc fragment (negative control). After 1.5 hr, PGS was added and samples were agitated for another 1.5 hr at 4°C. Finally, PGS was washed three times in lysis buffer

and bound proteins were eluted by boiling the PGS pellet for 5 min in 40 μl 2x SDS sample buffer. Samples were then resolved on a NuPage 4–12% Bis-Tris-Gradient Gel (Invitrogen), alongside mG6b-B-Fc (additional negative control) and stained with colloidal coomassie. Bands appearing in the mG6b-B-Fc pulldown, but not in the negative controls, were excised and subjected to mass spectrometry analysis (Orbitrap, Thermo Fisher Scientific, Paisley, UK). Corresponding areas from the control pulldown were cut and analyzed in parallel to account for background signals.

## In vitro binding assay

Nunc MaxiSorp plates (Thermo Fisher Scientific) were coated overnight with 50 μl of substrates, diluted in PBS (supplemented with 0.9 mM $CaCl_2$ and 0.5 mM $MgCl_2$ for laminins) at a concentration of 5 μg/ml. Plates were washed three times with Tris buffered saline (TBS) containing 0.1% Tween 20 (TBS-T) and blocked for 1.5 hr at 37°C with 2% fat free milk in TBS and 0.02% Tween 20. For heparin immobilization, biotinylated heparin (5 μg/ml) was added to streptavidin-coated wells for 1 hr at RT prior to the blocking step. After one washing step, recombinant G6b-B-Fc or human IgG-Fc fragment (negative control) in 3% BSA in TBS-T was added and incubated for 2 hr at 37°C. In competition assays, this incubation step was performed in the presence of the indicated compound. After five washing steps, wells were incubated with HRP-conjugated anti-human IgG antibody for 1 hr at RT at low agitation. Alternatively, monomeric, untagged G6b-B was incubated with anti-G6b-B antibody, and bound complexes were detected with HRP-conjugated anti-mouse IgG antibody. Plates were washed seven times and the signals were developed with TMB. The reaction was stopped by the addition of 2 M $H_2SO_4$ (50 μl/well) and absorbance at 450 nm and 570 nm (background) was measured with a Versa max plate reader (Molecular Devices, Wokingham, UK).

## Genome-wide cell-based genetic screening

The cell-based genome-wide genetic screen was performed essentially as described (*Sharma et al., 2018*). In brief, $3 \times 10^7$ Cas9-expressing HEK293 cells were transduced with a library of lentiviruses, each encoding a single gRNA from a pool of 90,709 individual gRNAs targeting 18,009 human genes at a low multiplicity of infection of 0.3 to increase the chances that each cell received a single gRNA. Ten million lentivirally transduced cells were selected using a blue fluorescent protein (BFP) marker three days after transduction using fluorescence-activated cell sorting. The sorted cells were placed back into culture and further selected for five days with 2 μg/mL puromycin. On day nine post transduction, $100 \times 10^6$ cells were stained with a recombinant protein consisting of the entire ectodomain of biotinylated human G6b-B clustered around phycoerythrin (PE)-conjugated streptavidin for an hour at room temperature. The cells were sorted using an XDP flow sorter and the $BFP^+$/$PE^-$ population collected, representing ~1% of the total cell population. A total of 600,000 cells were collected from which genomic DNA was extracted, and gRNA sequences were amplified by PCR before their abundances were determined by next generation sequencing. The enrichment of gRNA sequences targeting specific genes in the sorted versus unsorted populations were quantified from the sequence data using MAGeCK software (*Li et al., 2014*) as previously described (*Sharma et al., 2018*).

FGFRL1 was used as a positive control for testing the KO of SLC35B2, as it is known to interact with heparan sulfate (*Trueb, 2011*). Both G6b-B and FGFRL1 were produced as biotinylated proteins in HEK293 cells by co-transfection with a plasmid encoding a secreted form of the *Escherichia coli* biotin ligase, BirA (*Bushell et al., 2008*). The ectodomain of FGFRL1, aa 1–378, was subcloned from the Origene plasmid sc123844 by PCR with KOD enzyme and primers containing the Not-AscI restriction sites, and cloned into an expression vector containing a rat Cd4 domains 3 and 4 (CD3+4) tag and biotinylatable sequence. Avid fluorescent binding forms of the proteins were generated by conjugating the biotinylated recombinant ectodomain of FGFRL1 with streptavidin-PE. Parental and cells in which SLC35B2 was targeted were stained with the fluorescent reagent as described before (*Sharma et al., 2018*).

## Surface plasmon resonance

The interaction of the recombinant heterodimeric ('monomer') and homodimeric ('dimer') human G6b-B extracellular domain with different ligands was quantified using a BIAcoreTM 8K instrument (GE Healthcare, Little Chalfont, UK). Recombinant G6b-B proteins were immobilized on CM5 sensor

chips (GE Healthcare) via an Fc antibody using the Human Antibody Capture Kit (GE Healthcare). Immobilization levels ranged from 7800 to 9000 response units (RU) for the Fc antibody and 3000 to 4000 RU for the G6b-B proteins. Single cycle kinetics (SCK) measurements were undertaken with perlecan, heparin, fractionated HS and dp12. The analytes were injected in increasing concentrations of 0.1, 1, 10, 100 and 1000 nM. Analytes were flowed over the immobilized G6b-B surface at 30 µl/min with 60 s injection time and 60 s dissociation per concentration. In the 'reversed configuration', biotinylated heparin, HS and perlecan were immobilized on streptavidin sensor chips (GE Healthcare); fractionated HS and perlecan were biotinylated using the Lightning-Link Biotinylation kit (Innova Biosciences, Cambridge, UK). Immobilization levels of the biotinylated species were between 900 and 1000 RU. The SCK of 'monomeric' and 'dimeric' G6b-B were evaluated at 0.05, 0.5, 5, 50 and 500 nM. The analytes were flowed over the immobilized peptides at 10 µl/min with 180 s injection time and 360 s dissociation at each concentration. Data were collected from two replicates per experiment type and analyzed using the BIA evaluation software (GE Healthcare). Sensorgrams were double referenced prior to global fitting the SCK sensorgrams to one-to-one binding models in order to determine the rate constant of association ($k_{on}$) and dissociation ($k_{off}$). Binding affinities ($K_D$) were calculated from the equation $KD = k_{off}/k_{on}$.

## Theoretical modeling of G6b-B structure

The G6b-B ectodomain model was generated by submitting the amino acid sequence for G6b-B residues 18–142 to the RaptorX Structure Prediction server (http://raptorx.uchicago.edu/) (*Källberg et al., 2012*). Subsequent modeling of the K54D, K58D, R60E, R61E G6b-B mutants and all molecular graphics figure generation was carried out using PyMOL (The PyMOL Molecular Graphics System, Version 2.0 Schrödinger, LLC.). The electrostatic surfaces of both wild-type and mutant G6b-B models were calculated using the APBS suite (*Jurrus et al., 2018*).

## Crystallography

*Production of recombinant G6b-B and the anti-G6b-B Fab fragment.* The G6b-B extracellular domain (ECD) construct encompassing residues 18–133 which include the mutations N32D, S67A, S68A, S69A, T71A was expressed in mammalian HEK293 cells and purified by cation exchange and size-exclusion chromatography. The N32D mutation was used to remove the single potential N-linked glycosylation site that we showed by SDS-PAGE and mass spectrometry to be partially utilized. Intact mass spectrometry also revealed that after having made the N32D mutation, the measured mass of the protein was 948 Da greater than expected, a mass that is consistent with a common O-linked oligosaccharide, 2x sialic acid, galactose, N-acetyl galactosamine. The O-glycosylation was located using TOF-mass spectrometry of a chymotryptic digest of a region encompassing residues 66–80 with the sequence ASSSGTPTVPPLQPF. Each of the five potential sites was mutated to Ala individually in tandem with N32D. This showed that the predominant site was T73, although other Ser or Thr residues could be modified to a lesser degree. Finally, two constructs were made and tested for crystallization, one with four residues mutated to Ala but leaving T73 and one with all five residues mutated to Ala. The construct with four changes crystallized more readily and enabled us to solve the structure. The recombinant anti-G6b-B Fab fragment was also produced in HEK cells, synthetic genes for light and heavy chains were obtained from Invitrogen GeneArt. The G6b-B ECD-Fab complex was formed by incubating the components together for 2 hr at room temperature with G6b-B ECD in a 1.5 molar excess, and the complex was subsequently purified by size-exclusion chromatography. Protein was concentrated to 12 mg/ml in 20 mM Hepes (pH 7.1) and 75 mM NaCl, and finally incubated with 2 mM (10-fold molar excess) of the heparin oligosaccharide dp12 for 1 hr at 4˚C prior to setting up the crystallization experiment.

*Production of crystals and solving of structure.* Crystals were grown by vapor diffusion at 20˚C in 50 mM MES (pH 6.2), 10% PEG 550MME, 5% v/v glycerol, and 50 mM CaCl$_2$, and appeared within 3 days. Crystals were harvested straight out of the growth drop and cryo-cooled in liquid nitrogen. X-ray diffraction data were collected at 100K on beamline I03 at Diamond Light Source and processed using XDS (*Kabsch, 2010*) and Aimless (*Evans and Murshudov, 2013*) via AutoPROC (*Vonrhein et al., 2011*). The crystal was in space group C2 with the cell dimensions of a = 183.80 Å, b = 72.34 Å, c = 131.04 Å, β = 124.52˚, and extended to 3.1 Å resolution (*Table 2*).

The structure was initially solved by molecular replacement using the program Phaser (*McCoy et al., 2007*) and with a model of the Fab fragment generated from the PDB structure 4K2U (*Chen et al., 2013*) as the search model. This resulted in the placement of two Fab molecules in the asymmetric unit (Phaser Z-score after translation search = 10.2). Examination of the resulting electron density maps showed substantial unmodeled density in the vicinity of the CDR regions of both Fab molecules, which were interpreted as bound G6b-B ECD. Multiple rounds of model building in Coot (*Emsley and Cowtan, 2004*) and refinement using Refmac5 (*Murshudov et al., 1997*) resulted in a model encompassing about 90% (101 out of 116 residues) of the of G6b-B ECD chain. Residual density at that stage was identified as a single molecule of dp12-bound heparin, with the density covering 8 of the 12 saccharide units in dp12.

The final model represents a complex of G6b-B ECD, dp12 and Fab fragment chains in the ratio 2:1:2, respectively. The refined structure of G6b-B ECD chain has observable electron density for residues Pro19 to Thr38, Arg43 to Arg83 and Ile91 to Cys129. The G6b-B ECD, as expected, is shown to be a member of the IgV superfamily, with the solved structure comprising two antiparallel β-sheets formed by strands ABDE and A'CC'FG. There is also evidence from the electron density for O-linked glycosylation at Thr73 in both copies of the G6b-B ECD. Final refinement statistics for the G6b-B ECD-dp12-Fab dimer complex are given in *Table 2*.

## Size chromatography of G6b-B ECD

The G6b-B ECD protein encompassing residues 18–133 (N32D, S67A, S68A, S69A, T71A) was either analyzed immediately, or after incubation with heparin oligosaccharide dp12. A Superdex 75 10/300 GL column (GE Healthcare) was both equilibrated and run in 20 mM Hepes (pH 7.1) and 75 mM NaCl. dp12 was added to the G6b-B ECD at a 4-fold molar excess (150 µM final concentration). After the addition of dp12, the sample was aspirated gently and incubated for 90 min on ice, prior to SEC analysis. Columns were run at 0.3 ml/min, and 400 µl of G6b-B ECD samples were loaded (200 µg). A calibration curve was prepared in the same buffer using conalbumin (75 kDa), ovalbumin (44 kDa), carbonic anhydrase (29 kDa), ribonuclease A (13.7 kDa) and aprotinin (6.6 kDa) (LMM gel filtration standard kit, GE Healthcare). This calibration curve was then used to estimate the molecular weight of both G6b-B ECD and G6b-B ECD +dp12 in order to determine their polymeric states.

## Flow-cytometric analysis of heparin binding in transfected CHO cells

A5 CHO cells were kindly provided by Ana Kasirer-Friede and Sanford Shattil (University of California, San Diego). A test for mycoplasma contamination was negative. Transfections of WT or mutant hG6b-B into CHO cells were carried out in 6-well plates ($3 \times 10^5$ cells in 2 ml DMEM medium, supplemented with 10% fetal bovine serum, 2 mM glutamin) using polyethylenimine (Sigma-Aldrich) as described (*Ehrhardt et al., 2006*). Cells were harvested 2 days after transfection by detaching them with accutase, and resuspended in PBS containing 0.2 mg/ml BSA and 0.02% sodium azide. Cells were incubated with heparin-biotin (10 µg/ml) and mouse anti-human G6b-B antibody for 45 min at RT, washed twice, and incubated with streptavidin-PE (BD Biosciences) and anti-mouse-alexa488 antibody (Invitrogen). Cells were fixed with 1% formaldehyde and analyzed on a BD FACSCalibur (BD Biosciences).

## Aggregometry

Platelet rich plasma (PRP) was prepared from blood collected from healthy drug-free volunteers as described previously (*Dawood et al., 2007*). Donors gave full informed consent according to the Helsinki declaration. Ethical approval for collecting blood was granted by Birmingham University Internal Ethical Review (ERN_11–0175 and ERN_15–0973). In brief, 9 volumes of blood were collected into 1 vol of 4% (w/v) sodium citrate solution. Blood was centrifuged at 200 × g for 20 min at RT and PRP was collected. Platelet aggregation was measured using a lumi-aggregometer (Chrono-Log, Abingdon on Thames, UK, Model 700).

## Platelet adhesion assay

This assay was performed as described previously (*Bellavite et al., 1994*). In brief, Nunc MaxiSorp plates were coated overnight with 50 µl of substrates, diluted in PBS at a concentration of 10 µg/ml, except for collagen which was used at 2.5 µg/ml. Plates were then washed three times and blocked

with 2% BSA in PBS for 1 hr at 37°C. After washing, 50 µl heparinase III (5 mU/ml) or buffer (20 mM Tris-HCl (pH 7.5), 0.1 mg/ml BSA and 4 mM $CaCl_2$) were added to each well and incubated for 1.5 hr. After three washing steps, 50 µl of platelet suspension modified Tyrode's buffer, prepared as previously described (*Pearce et al., 2004*), at a concentration of $1 \times 10^8$/ml were added and incubated for 1 hr at 37°C. After three washing steps with PBS, 140 µl of substrate solution was added to each well and incubated on a rocker at RT for 40 min. Then, 50 µl of 3M NaOH was added and the signal was quantified 5 min later by measuring the absorbance at 405 nm and 620 nm (background). Percentage of adhesion was calculated by normalizing the measured ODs to the signal obtained by directly lysing 50 µl of platelet suspension.

## Flow cytometric analysis of platelet activation

5 µl staining solution, containing 1.5 µg fibrinogen-Alexa488 conjugate (Invitrogen) and 1 µg of anti-TLT-1-Alexa 647 antibody (Biotechne, Abingdon, UK) and 5 µl of whole blood were provided in a well of a 96-well plate. Stimulation was started by adding 40 µl of heparin, APAC (0.05 µM final concentration) or buffer, with or without CLEC-2 antibody (3 µg/ml final concentration; Bio-Rad, Oxford, UK). The plate was incubated in the dark for the indicated time and the reaction was stopped by addition of 200 µl 1% ice-cold formalin. Samples were analyzed on a BD Accuri flow cytometer. Platelets were gated using forward and side scatter.

## Preparation and culture of mouse megakaryocytes

Megakaryocytes were prepared as previously described (*Dumon et al., 2006*; *Mazharian et al., 2011*). In brief, bone marrow cells were obtained from mouse femurs and tibias by flushing, and cells expressing lineage-specific surface markers (CD16/CD32, Gr1, B220, or CD11b) were depleted using immunomagnetic beads (sheep anti-rat IgG Dynabeads, Invitrogen). The remaining population was cultured in 2.6% serum-supplemented StemPro medium with 2 mM L-glutamine, penicillin/streptomycin, and 20 ng/mL of murine stem cell factor at 37°C under 5% $CO_2$ for 2 days, and for a further 4 days in the presence of stem cell factor and 50 ng/mL thrombopoietin (37°C, 5% $CO_2$). Mature megakaryocytes were then enriched using a 1.5%/3% bovine serum albumin gradient under gravity (1 g) for 45 min at room temperature.

## Microscopical analysis of platelet and MK adhesion

Glass Coverslips (5 mm diameter) were incubated with 50 µl of perlecan (25 µg/ml), fibrinogen (25 µg/ml) or both overnight at 4°C. Surfaces were then blocked with denatured BSA (5 mg/ml) for 1 hr at room temperature. After washing, 50 µl heparinase III (12.5 mU/ml) or buffer were added to each well and incubated for 1.5 hr at 37°C. Platelets ($2 \times 10^7$/ml, 50 µl) were transferred to the slides and incubated at 37°C for 45 min in a humid atmosphere. Mature megakaryocytes ($6 \times 10^3$/ml, 100 µl) were incubated for 5 hr. Non-adherent cells were removed by gently washing wells with PBS and adherent cells were fixed with 3.7% paraformaldehyde and permeabilized with 0.2% Triton-X 100 in water. MKs were stained with tubulin-antibody for 1 hr followed by anti-mouse-Alexa-488 and rhodamin-conjugated phalloidin for 30 min, and coverslips were mounted onto microscope slides for imaging using or Antifade Mountant with DAPI (Invitrogen). Platelets were stained with phalloidin-Alexa-488 for 1 hr and coverslips were mounted using Hydromount (National Diagnostics, Nottingham, UK). Images were captured by a Zeiss Axio Observer.Z1/7 epifluorescence microscope using ZEN Software and 20x (MK) or 63x oil immersion (platelet) plan apochromat objectives.

For platelets, each coverslip was imaged in three random areas. For analysis, the central quarter of each field of view was cropped ($1024 \times 1024$ pixels) and ilastik pixel classifier software was used to outline a binary segmentation (*Sommer et al., 2011*). To distinguish touching platelets, KNIME analytic platform was used to identify the centre of individual platelets manually (*Berthold et al., 2009*). These coordinates were used to produce the final segmentation of individual platelets, and platelet size was subsequently calculated.

For MK, three tiles of $3 \times 3$ images were acquired per coverslip. Average surface area per cell was calculated by analyzing total surface area and number of cells per image by using ImageJ. Both, imaging and analysis were performed blinded.

## Western blotting and immunoprecipitation

Washed human or mouse platelets ($5 \times 10^8$/ml) in the presence of 10 μM integrilin or lotrafiban (integrin αIIbβ3 inhibitors), respectively, were incubated with the respective compound under stirring conditions (1200 rpm, 37°C) for the indicated time. Platelets were lysed by the addition an equal volume of ice cold 2 x lysis buffer and insoluble cell debris was removed by centrifugation for 10 min at 13,000 x g, at 4°C.

For immunoprecipitations, whole cell lysates (WCLs) were precleared using protein A Sepharose (Sigma-Aldrich) for 30 min at 4°C. G6b-B was immunoprecipitated from collagen-WCLs with anti-G6b-B antibody and protein A sepharose overnight at 4°C as previously described (*Mazharian et al., 2012*).

WCLs were either boiled in SDS-loading buffer and analyzed by SDS-PAGE (NuPage 4–12% Bis-Tris-Gradient Gel) and traditional western blotting or, for quantitative analysis, analyzed with an automated capillary-based immunoassay platform (Wes, ProteinSimple, San Jose, USA), according to the manufacturer's instructions. Briefly, WCLs were diluted to the required concentration with 0.1X sample buffer, then prepared by adding 5X master mix containing 200 mM dithiothreitol (DTT), $5 \times$ sample buffer and fluorescent standards (Standard Pack 1, PS-ST01-8) and boiled for 5 min at 95°C. Samples, antibody diluent 2, primary antibodies and anti-rabbit secondary antibody, together with luminol S-peroxide mix and wash buffer, were displaced into Wes 12–230 kDa prefilled microplates (pre-filled with separation matrix 2, stacking matrix 2, split running buffer two and matrix removal buffer, SM-W004). The microplate was centrifuged for 5 min at 2500 rpm at room temperature. To start the assays, the capillary cartridge was inserted into the cartridge holder and the microplate placed on the plate holder. To operate Wes and to analyze results, Compass Software for Simple Western was used (version 3.1.7, ProteinSimple). Separation time was set to 31 min, stacking loading time to 21 s and sample loading time to 9 s. Primary antibodies were incubated for 60 min and the High Dynamic Range (HDR) profile was used for detection. For each antibody, a lysate dilution experiment was performed first to confirm the optimal dynamic range of the corresponding protein on Wes. This was followed by an antibody optimization experiment to compare a range of dilutions and to select an antibody concentration that was close to saturation level to allow a quantitative comparison of signals between samples. The optimized antibody dilutions and final lysate concentrations were as indicated in the key resources table.

## Statistical analysis

All data are presented as mean ± standard deviation (SD). Statistical significance was analyzed by one-way or two-way ANOVA, followed by the appropriate post hoc test, as indicated in the figure legend, using GraphPad Prism 6 (GraphPad Software Inc, San Diego, CA, USA).

# Acknowledgements

The authors would like to thank all of the voluntary blood donors and the phlebotomists as well as Jamie Webster from the University of Birmingham Core Protein Expression Facility and Silke Heising for excellent technical assistance, Jeremy A Pike for assistance with image and KNIME analysis, the Advanced Mass Spectrometry Facility, and all members of the Biomedical Services Unit for exceptional maintenance of the mouse colonies.

# Additional information

### Competing interests

Jordan Lane: was an employee at Sygnature Discovery Limited at the time of the study, performing surface plasmon resonance experiments as part of a paid service. Scott Pollack: is an employee at Sygnature Discovery Limited, performing surface plasmon resonance experiments as part of a paid service. Riitta Lassila: is CSO and shareholder of Aplagon Oy, Helsinki, Finland. Annukka Jouppila: receives research funding from Aplagon Oy, Helsinki, Finland. Derek J Ogg, Tina D Howard, Helen J McMiken, Juli Warwicker, Catherine Geh: is an employee at Peak proteins Limited, performing crystallography and protein expression studies as part of a paid service. Rachel Rowlinson: is employee

at Peak proteins Limited, performing crystallography and protein expression studies as part of a paid service. W Mark Abbott: is CEO of Peak proteins Limited, performing crystallography and protein expression studies as part of a paid service. The other authors declare that no competing interests exist.

## Funding

| Funder | Grant reference number | Author |
|---|---|---|
| British Heart Foundation | RG/15/13/31673 | Jun Mori<br>Zoltan Nagy<br>Alexandra Mazharian<br>Yotis A Senis |
| British Heart Foundation | FS/13/1/29894 | Yotis A Senis |
| Deutsche Forschungsgemeinschaft | VO 2134-1/1 | Timo Vögtle |
| Medical Research Council | Confidence in Concept 2018 | Timo Vögtle<br>Yotis A Senis |
| Wellcome | 206194 | Gavin J Wright |
| British Heart Foundation | FS/15/58/31784 | Alexandra Mazharian |
| Agence Nationale de la Recherche | ANR-17-CE14-0001-01 | Anita Eckly |

The funders had no role in study design, data collection and interpretation, or the decision to submit the work for publication.

## Author contributions

Timo Vögtle, Formal analysis, Funding acquisition, Validation, Investigation, Methodology, Writing—original draft, Writing—review and editing; Sumana Sharma, Formal analysis, Investigation, Methodology, Writing—review and editing; Jun Mori, Formal analysis, Funding acquisition, Investigation, Methodology, Writing—review and editing; Zoltan Nagy, Formal analysis, Investigation, Methodology, Writing—original draft, Writing—review and editing; Daniela Semeniak, Cyril Scandola, Mitchell J Geer, Juli Warwicker, Anita Eckly, Michael R Douglas, Investigation, Writing—review and editing; Christopher W Smith, Helen J McMiken, Catherine Geh, Rachel Rowlinson, Investigation; Jordan Lane, Scott Pollack, Investigation, Methodology, Writing—original draft; Riitta Lassila, Annukka Jouppila, Resources, Writing—review and editing; Alastair J Barr, Methodology, Writing—review and editing; Derek J Ogg, Data curation, Formal analysis, Investigation, Methodology, Writing—review and editing; Tina D Howard, Supervision, Investigation; W Mark Abbott, Formal analysis, Supervision, Methodology; Harald Schulze, Formal analysis, Writing—review and editing; Gavin J Wright, Formal analysis, Funding acquisition, Investigation, Methodology, Writing—original draft, Writing—review and editing; Alexandra Mazharian, Formal analysis, Supervision, Funding acquisition, Investigation, Methodology, Writing—review and editing; Klaus Fütterer, Formal analysis, Validation, Investigation, Methodology, Writing—original draft; Sundaresan Rajesh, Formal analysis, Investigation; Yotis A Senis, Conceptualization, Resources, Formal analysis, Supervision, Funding acquisition, Validation, Investigation, Methodology, Writing—original draft, Project administration, Writing—review and editing

## Author ORCIDs

Timo Vögtle https://orcid.org/0000-0002-9400-4701
Sumana Sharma https://orcid.org/0000-0003-0598-2181
Jun Mori http://orcid.org/0000-0002-6212-1604
Zoltan Nagy https://orcid.org/0000-0001-6517-2071
Mitchell J Geer https://orcid.org/0000-0003-1457-987X
Scott Pollack https://orcid.org/0000-0002-8176-0997
Riitta Lassila https://orcid.org/0000-0002-1911-3094
Alastair J Barr http://orcid.org/0000-0001-7738-8419

Derek J Ogg ⓘ https://orcid.org/0000-0002-7751-5913
Harald Schulze ⓘ https://orcid.org/0000-0003-1285-6407
Gavin J Wright ⓘ http://orcid.org/0000-0003-0537-0863
Alexandra Mazharian ⓘ https://orcid.org/0000-0002-0204-3325
Klaus Fütterer ⓘ https://orcid.org/0000-0001-7445-5372
Yotis A Senis ⓘ https://orcid.org/0000-0002-0947-9957

### Ethics

Human subjects: Blood was collected from healthy drug-free volunteers. Donors gave full informed consent according to the Helsinki declaration. Ethical approval for collecting blood was granted by Birmingham University Internal Ethical Review (ERN_11-0175 and ERN_15-0973).
Animal experimentation: All animal procedures were undertaken with the U.K. Home Office approval (project license No P46252127) in accordance with the Animals (Scientific Procedures) Act of 1986.

### Decision letter and Author response

Decision letter https://doi.org/10.7554/eLife.46840.059
Author response https://doi.org/10.7554/eLife.46840.060

## Additional files

### Supplementary files

• Transparent reporting form
DOI: https://doi.org/10.7554/eLife.46840.041

### Data availability

Diffraction data have been deposited in PDB under the accession code 6R0X.

The following dataset was generated:

| Author(s) | Year | Dataset title | Dataset URL | Database and Identifier |
|---|---|---|---|---|
| Timo Vögtle, Sumana Sharma, Jun Mori, Zoltan Nagy, Daniela Semeniak, Cyril Scandola, Mitchell J Geer, Christopher W Smith, Jordan Lane, Scott Pollack, Riitta Lassila, Annukka Jouppila, Alastair J Barr, Derek J Ogg, Tina D Howard, Helen J McMiken, Juli Warwicker, Catherine Geh, Rachel Rowlinson, W Mark Abbott, Anita Eckly, Harald Schulze, Gavin J Wright, Alexandra Mazharian, Klaus Fütterer, Sundaresan Rajesh, Michael R Douglas, Yotis A Senis | 2019 | G6b-B in complex with dp12 | http://www.rcsb.org/structure/6R0X | Protein Data Bank, 6R0X |

The following previously published datasets were used:

| Author(s) | Year | Dataset title | Dataset URL | Database and Identifier |
|---|---|---|---|---|
| Cai Z, Yarovoi S V, Zhu Z, Rauova L, | 2015 | Crystal structure of platelet factor 4 complexed with fondaparinux | https://www.rcsb.org/structure/4R9W | Protein Data Bank, 4R9W |

| Hayes V, Lebedeva T, Liu Q, Poncz M, Arepally G, Cines D B, Greene M I | | | | |
|---|---|---|---|---|
| Dahms S O, Mayer M C, Roeser D, Multhaup G, Than M E | 2015 | X-ray structure of the amyloid precursor protein-like protein 1 (aplp1) e2 domain in complex with a heparin dodecasaccharide | https://www.rcsb.org/structure/4RDA | Protein Data Bank, 4RDA |
| Fukuhara N, Howitt J A, Hussain S A, Hohenester E | 2008 | Drosophila Robo IG1-2 (monoclinic form) | https://www.rcsb.org/structure/2VRA | Protein Data Bank, 2VRA |
| Pellegrini L, Burke D F, von Delft F, Mulloy B, Blundell T L | 2000 | Crystal structure of a ternary fgf1-fgfr2-heparin complex | https://www.rcsb.org/structure/1E0O | Protein Data Bank, 1E0O |
| Schlessinger J, Plotnikov A N, Ibrahimi O A, Eliseenkova A V, Yeh B K, Yayon A, Linhardt R J, Mohammadi M | 2000 | Crystal structure of a ternary fgf2-fgfr1-heparin complex | https://www.rcsb.org/structure/1FQ9 | Protein Data Bank, 1FQ9 |

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
