## [Decision Letter]

Thank you for submitting your article "Heparan sulfates are critical regulators of the inhibitory megakaryocyte-platelet receptor G6b-B" for consideration by *eLife*. Your article has been reviewed by Philip Cole as the Senior Editor, a Reviewing Editor, and three reviewers. The reviewers have opted to remain anonymous.

The reviewers have discussed the reviews with one another and the Reviewing Editor has drafted this decision to help you prepare a revised submission.

Summary:

This manuscript reports on novel aspects of the biology of G6b-B, a receptor highly expressed in platelets and megakaryocytes. The authors provide a strong rationale for the biomedical relevance of G6b-B: its loss or absence has been shown to result in thrombocytopenia, platelet dysfunction, and bleeding in mice and humans. The authors' had a compelling goal for this work: to identify the physiologic ligand for G6b-B, which has previously been unknown. Vӧgtle et al., provide strong evidence that the heparan sulfate glycosaminoglycan (HS-GAG) is the ligand of the G6b-B receptor and provide both structural and functional evidence of the involvement of this interaction in platelet function through the induction of intracellular ITIM-associated downstream signaling by the tyrosine phosphatases Shp1 and Shp2. Strengths of this paper include the use of a diversity of methods to probe the structure and function of G6b-B, including proteomics, a CRISPR genetic screen, crystallography, and in vitro biochemical studies.

The manuscript is well written with regards to the in vitro ligand interaction and structural data. However, this paper would strongly benefit from expanding upon the physiological relevance of the G6b-B-HS/perlecan interaction in bone marrow resident megakaryocytes and in platelets. In addition, in a number of instances the claims made in this manuscript are not adequately supported by the data.

Essential revisions:

1) The sulfation pattern within the GAG-domains, as well as their length and spacing, contribute to the structural heterogeneity of HS and create specific binding sites for protein ligands. The distribution, length, and modification level of such domains appear to be tightly regulated in a tissue/cell-specific fashion suggesting that the regulatory role of HS occurs in a spatiotemporal manner by interacting with unique arrays of protein ligands in different tissues and at different developmental/pathological stages. The structural data show Heparin binding to G6b-B. Biological assays use perlecan, heparin and APAC. Heparin and perlecan do not bind tightly to platelets, whereas APAC does. These molecules differ substantially in HS sequence and structure. The authors do not address the differences. Also, heavy O-glycosylation of G6b-B could affect binding. This has to be considered and at least discussed in the paper.

2) Figure 1: Femur bone marrow tissue sections should be included in immunohistochemical analyses of G6b-B ligand expression. Overview sections would be helpful in all organs.

3) Figure 8: Can the finding in A be replicated under flow conditions? It would be interesting to test spreading using human platelets. The immunofluorescent micrographs in figure 8C must include more than 2 platelets per field to be convincing. The number of cells analyzed in the figure legend of the spreading should be quantified (Figure 8C).

4) Given the mass spec data described in Table 1, why was laminin excluded from the adhesion assays of Figure 8? Moreover, what is the explanation for heparinase treatment increasing the adhesion of G6b-/- mouse platelets (figure 8B) and also affecting their spreading (Figure 8C)? How might HS have an effect on these knockout platelets? Is there another receptor on platelets binding to the backbone of perlecan?

5) Is there a difference in glycosylation of the G6b-B receptor between megakaryocytes and platelets, which may result in profound differences in binding affinities? Data on the G6b-B-HS/perlecan interaction in megakaryocyte function are lacking, especially given the emphasis on this point in the introduction and Discussion section. The conclusion that megakaryocytes are activated by the interaction is hard to draw from the presented results. Megakaryocytes could be always active in the bone marrow. And how might this translate into the defects seen in G6b-B-deficient platelets?

6) Figure 9: The context of megakaryocyte G6b-B interactions is an important point and yet barely identifiable by the micrographs of Figure 9A,B. It is not clear which mouse phenotype is shown in Figure 9A and B. If only control samples are shown, results from both phenotypes should be demonstrated. It is unclear if there are differences in megakaryocyte number or distribution. Overview data of bone marrows would be helpful. In subsection “Biological effects of perlecan, heparin and HS on platelets and MKs” the sentence: "we found very few WT and G6b-/- MKs adhered to perlecan-coated surface formation", is unclear. Did the authors quantify the distance between perlecan surfaces and megakaryocytes in the entire femur sections? Again, overview sections and quantification would be helpful. The authors show differences in megakaryocyte size between WT and KO mice when cultured in the presence of different substrates. This difference could be due to different stages of maturation and differentiation, as the images suggest. Is there any difference in the maturation of megakaryocyte between phenotypes?

7) In Figure 9C, number of cells quantified would again be good information to have in the figure legend. Axes should be added to aggregation graphs. Control (with no stimulation) should be shown. The data can be consolidated in a more palatable graph. Does plasma have an impact on HS binding to G6b-B?

8) The authors observe that perlecan does not seem to have an effect on either platelet aggregation (Figure 10) or G6b-B phosphorylation (Figure 12), likely due to the need for the proteoglycan to be immobilized on a surface, rather than in suspension. The effects of perlecan on platelet G6b-B signaling is a major component of this paper and should be elucidated by allowing platelets to adhere on immobilized perlecan prior to lysis and blotting for phosphotyrosine. The conclusion that G6b-B and its ligand perlecan inhibit activation in platelets is not really supported by direct evidence. The evidence is based on APAC binding to platelets, hence more direct evidence would be helpful.

9) The text corresponding to Figure 11 and Figure 12 needs to be streamlined, as it is hard to follow. The figures could be edited/consolidated for clarification.

10) Were data in Figure 11 obtained with washed platelets or platelets in plasma? Signaling is not induced by soluble perlecan. Can the authors immobilize perlecan to test if immobilized perlecan induces signaling?

11) In Figure 4—figure supplement 1, why do CHO cells transfected with G6b-B mutant bind less efficiently than mock transfected cells? One can suppose a dominant negative effect? CHO cells constitutively express G6b-B?

12) In Figure 8, given the important reduction in platelet binding onto fibrinogen and collagen coated surfaces in the presence of intact immobilized perlecan and given the percentage of adhesion, one would expect that, after removing HS from perlecan in the same settings, platelet adhesion to both ECMs should increase in an additive manner above the level of the two ECMs alone. This is further supported by the previous finding of Lord et al., 2011 showing that plaletet binding to C-terminal fragment of perlecan, endorepellin, enhances collagen-mediated platelet responses. Please comment.

13) The results shown in Figure 8B showed a negligible interaction of both WT and G6b-/- platelets to perlecan, this raises the doubt of the significance of the experiments shown in Figure 8C. Further, the differences in platelet area might be due to the intrinsic differences in platelet volume previously demonstrated by the same authors. Further, it is not clear how removing the HS and thus exposing the core protein of perlecan to platelet a2b1 platelet spreading should decrease.

14) There is a lack of correspondence between the text and the panels shown in Figure 9. Although potentially interesting, the data on megakaryocytes appear too preliminary and do not add significance to the work.

15) The data on the opposite effects of soluble and conjugated heparin on platelet function are interesting. The authors concluded that these are due to a differential engagement of G6b-B and related downstream related signaling. To definitely prove the fundamental relevance of the SHPs-related inhibitory signaling, the authors should perform the experiments on platelet function with SHP1/2 KO platelets and by using SHPs specific inhibitors.

Concerns about interpretation:

1) While the authors demonstrate that perlecan is a ligand for G6b-B, their claim that this is the physiologically relevant ligand is not adequately supported. Given that prior work from mice and humans has demonstrated that the phenotype of reduced/absent G6b-B includes (1) decreased in vitro platelet aggregation/activation and (2) thrombocytopenia, it would seem logical to attempt to identify the ligand in lysates from tissues that would be relevant for those phenotypes. This would correspond to (1) blood/plasma (where platelets would be activated) or (2) bone marrow (where platelets are produced). Instead the authors looked for a G6b-B ligand in vena cava lysates; no rationale was provided as to the relevance of this to the human or mouse G6b-B knock out phenotypes. In fact, the authors note that their identified ligand perlecan would be inaccessible to platelets in intact blood vessels, making it hard to imagine that this would be a relevant ligand for platelet aggregation/activation.

2) Furthermore, additional information is needed to validate perlecan as a hit of the immunoprecipitation proteomics experiment. The hits from the G6b-B pulldown are listed in order of a "score" in Table 1, but it is not clear how the authors determined or defined this score. The authors mention that proteins that were "prominently present in the negative control" were not included in their list. It would be preferable if they could provide a complete data set for their proteomics results, and calculate a fold enrichment of the identified proteins/peptides in their experimental vs control condition. A reverse immunoprecipitation experiment (pulling down perlecan with an antibody and seeing if G6b-B is pulled down) would be a very useful experiment to confirm these findings.

3) Further experimental detail is also needed in order to support the authors' claim that heparan sulfate and not chondroitin sulfate mediates binding of G6b-B to perlecan. The CRISPR knock out screen is suggestive, as it resulted in a number of hits in genes in the heparan sulfate biosynthesis pathway. However, only one hit from this screen was specifically validated by flow cytometry. It would be useful to validate a few more of the hits by making targeted CRISPR knock outs, and specifically to individually target EXTL3 and CSGALNACT_1/2_ (which are the enzymes that commit tetrasaccharide precursors to either heparan sulfate or chondroitin sulfate), and then look for G6b-B binding to the knock down cells by flow cytometry as in Figure 3C. It would also be helpful to generate a perlecan CRISPR HEK293 knock out and test for G6b-B binding by flow.

4) Finally, the functional data that are provided here lead the authors to conclude that the function of G6b-B is to inhibit platelet activation in the presence of perlecan heparan sulfate. Given that the phenotype of decreased/absent G6b-B is bleeding and decreased platelet activation, it is surprising that the in vitro experiments suggest that absence of G6b-B would result in excessive platelet activation. This seeming contradiction calls in to question whether perlecan heparan sulfate is truly the physiologically pertinent ligand for the clinical phenotype.

---

## [Author Response]

*[…] The manuscript is well written with regards to the* in vitro *ligand interaction and structural data. However, this paper would strongly benefit from expanding upon the physiological relevance of the G6b-B-HS/perlecan interaction in bone marrow resident megakaryocytes and in platelets. In addition, in a number of instances the claims made in this manuscript are not adequately supported by the data.*

We want to thank the editors and the reviewers for critical evaluation of our manuscript. Please find our answers to the reviewer’s questions below. We have made substantial changes to the manuscript by incorporating new data and also by rephrasing the text and by providing additional information, as requested by the reviewers. We hope the reviewer feel that improved version of the manuscript is suitable for publication in *eLife*.

Essential revisions:1) The sulfation pattern within the GAG-domains, as well as their length and spacing, contribute to the structural heterogeneity of HS and create specific binding sites for protein ligands. The distribution, length, and modification level of such domains appear to be tightly regulated in a tissue/cell-specific fashion suggesting that the regulatory role of HS occurs in a spatiotemporal manner by interacting with unique arrays of protein ligands in different tissues and at different developmental/pathological stages. The structural data show Heparin binding to G6b-B. Biological assays use perlecan, heparin and APAC. Heparin and perlecan do not bind tightly to platelets, whereas APAC does. These molecules differ substantially in HS sequence and structure. The authors do not address the differences. Also, heavy O-glycosylation of G6b-B could affect binding. This has to be considered and at least discussed in the paper.

We thank the reviewer for highlighting these important points related to structural diversity of HS-GAGs that regulate ligand-receptor interactions and the possible implications in pathological settings. We have added two major paragraphs to this effect in the Discussion section.

Regarding the binding affinities of heparin, HS and perlecan to recombinant G6b-B, using SPR, we demonstrate that the binding affinities of heparin and HS are comparable, whereas perlecan is slightly lower. This may be due to structural differences in the perlecan associated HS chains, as raised by the reviewer.

Although G6b-B is O-glycosylated, our crystal structure shows that the glycosylation sites are all spatially well separated from the ligand binding surface, and are not likely to sterically impede ligand binding. In addition, all the recombinant G6b-B molecules used in this study were produced in a mammalian cell line and are glycosylated as well. Hence, we currently have no evidence that glycosylation would affect ligand binding. We have added this information to the Discussion section.

2) Figure 1: Femur bone marrow tissue sections should be included in immunohistochemical analyses of G6b-B ligand expression. Overview sections would be helpful in all organs.

Femur bone marrow sections have been added to Figure 1. Overview sections are shown in Figure 1—figure supplement 1 and Figure 1—figure supplement 2.

3) Figure 8: Can the finding in A be replicated under flow conditions? It would be interesting to test spreading using human platelets. The immunofluorescent micrographs in figure 8C must include more than 2 platelets per field to be convincing. The number of cells analyzed in the figure legend of the spreading should be quantified (Figure 8C).

Platelets do not adhere or spread well on perlecan, as originally described by Whitelock and co-workers (Lord et al., 2009). Thus, we struggled to observe any platelets on a perlecan-coated surface under various conditions tested. It is for this reason that we tried co-coating with perlecan and either fibrinogen or collagen. It is only when we treated perlecan with heparinase III that we observed more platelets adhering and spreading to a greater extent on various surfaces, fitting with our model that this is due to negative signalling via G6b-B.

For reasons given above, we have yet to optimize conditions for platelet flow adhesion studies, which will take a substantial amount of time. We are also starting to use purified perlecan and perlecan fragments provided by Prof Whitelock. Data is not ready to present at this stage, but will form the basis of a follow-up study.

We have included new human platelet spreading data (Figure 8C), demonstrating that human platelets adhere, but do not spread on perlecan. Heparinase III treatment increased surface area of platelets, but average size was much smaller than platelets spreading on fibrinogen, which served as a positive control.

Due to the low number of platelets adhering to perlecan, it is difficult to find more than two platelets in close proximity. We would need to use a lower magnification objective to show multiple platelets, which would result in a loss of structural details of the depicted platelets. Since we provide a quantitative analysis of platelet size, we kept the same representative immunofluorescent micrographs, to illustrate qualitative changes in platelets under different conditions. We also added dot plots for all cells analysed as a Figure 8—figure supplement 1.

We have added platelet numbers quantified to figure legend.

4) Given the mass spec data described in Table 1, why was laminin excluded from the adhesion assays of Figure 8? Moreover, what is the explanation for heparinase treatment increasing the adhesion of G6b-/- mouse platelets (figure 8B) and also affecting their spreading (Figure 8C)? How might HS have an effect on these knockout platelets? Is there another receptor on platelets binding to the backbone of perlecan?

Although laminins were identified in our proteomic analysis (Table 1), we did not detect any interaction between recombinant G6b-B and various forms of laminin in our in vitro binding assay (Figure 2A), thus we did not pursue further. We suspect the reason for the high laminin signal in our G6b-B pulldown is due to indirect association with laminin via perlecan, which was pulled down. It is well established that laminin interacts with perlecan (Battaglia et al., 1992).

Results from the adhesion and spreading experiments with WT and G6b-B KO platelets on perlecan (Figure 8 D) indeed suggests platelet binding to the perlecan core or HS side chains via other receptors. The obvious candidate for binding to the perlecan protein core is the integrin α2β1, which has previously been shown to bind to the C-terminus of perlecan, called endorepellin (Bix et al., 2007). However, we cannot exclude the possibility that platelets also bind to perlecan via other adhesion receptors, either directly, or indirect via perlecan-associated extracellular matrix proteins. The different spreading response of G6b-B KO platelets to perlecan with or without HS chains indeed raises the possibility that HS also binds to other platelet surface receptors, which may be masked in the presence of G6b-B. This is now discussed in the Results section and Discussion section of the revised manuscript.

5) Is there a difference in glycosylation of the G6b-B receptor between megakaryocytes and platelets, which may result in profound differences in binding affinities? Data on the G6b-B-HS/perlecan interaction in megakaryocyte function are lacking, especially given the emphasis on this point in the introduction and Discussion sections. The conclusion that megakaryocytes are activated by the interaction is hard to draw from the presented results. Megakaryocytes could be always active in the bone marrow. And how might this translate into the defects seen in G6b-B-deficient platelets?

We thank the reviewers for raising these excellent points. We currently have no evidence that G6b-B is differentially glycosylated in MKs and platelets. We previously showed that MK and platelet G6b-B migrate at the same molecular weight by Western blotting (Mazharian et al., 2012), suggesting G6b-B is not differentially glycosylated in MKs and platelets. However, further biochemical analysis is required to validate this observation. We have included this information in the Discussion section. Please also refer to our answer to point 1, above.

Although MK functional data is limited, it provides proof-of-principle that G6b-B is also inhibitory in MKs. This is the first time that perlecan has been shown to have a biological effect on MK function, which is mediated through G6b-B. These experiments were challenging due to the limited number of MKs that adhere to perlecan, requiring a substantial amount of optimization that we will now build on in future studies.

We have toned down the potential physiological implications of our MK findings. We agree that MKs may already be in an ‘activated’ state in the bone marrow, but that G6b-B helps maintain a basal level of activity. A hypothesis we are currently testing is whether contact between MK G6b-B and perlecan in the vessel wall polarizes the MK, providing directionality for proplatelet formation into the vessel lumen. This is now discussed in the revised manuscript.

6) Figure 9: The context of megakaryocyte G6b-B interactions is an important point and yet barely identifiable by the micrographs of Figure 9A,B. It is not clear which mouse phenotype is shown in Figure 9A and B. If only control samples are shown, results from both phenotypes should be demonstrated.

We thank the reviewers for raising these important points and apologize for the confusing labelling and description of the data in the original version of the figure. We have now addressed these points and also rephrased the text and rearranged the subfigures, making it easier to follow.

In the original Figure 9A,B, we had only shown staining of WT bone marrow sections. We have now added data from G6b KO mice and quantification of the number of MKs, as well as MK clustering, in bone marrow sections from both genotypes.

It is unclear if there are differences in megakaryocyte number or distribution. Overview data of bone marrows would be helpful.

This is an important point: We have now generated novel images taken over the complete mouse femur and stitched the images to get a complete section overview (Figure 9—figure supplement 1). This allowed us to identify MKs over the distinct bone areas and we can now conclude that the MK distribution is comparable between knockout and wildtype animals. Clearly, the number of MKs is significantly higher in G6b KOanimals compared to wildtype controls (Figure 9B). We have clarified this statement in the Results section of the revised manuscript.

In addition, we have added electron microscopy images of MKs in WT and *G6b* KO bone marrow (Figure 9D-E and Figure 9—figure supplement 2), complementing the data above. This data shows, that there is no significant difference in the maturation of MKs in G6b KO mice. In summary, findings demonstrate a profound increase in the number of MKs around perlecan-positive vessels in the bone marrow.

In subsection “Biological effects of perlecan, heparin and HS on platelets and MKs” the sentence: "we found very few WT and G6b-/- MKs adhered to perlecan-coated surface formation", is unclear. Did the authors quantify the distance between perlecan surfaces and megakaryocytes in the entire femur sections? Again, overview sections and quantification would be helpful.

Thank you for pointing this out. We have rephrased the result section and also moved the findings from our in vitro experiment, the seeding of ex vivo-differentiated MKs onto different surfaces (former Figure 9C) into a new Figure (Figure 10), to separate it more clearly from the microscopic analysis of bone marrow sections.

We have not quantified the distance between perlecan surfaces and megakaryocytes, since in our experience a quantification of this kind of interaction in an immunofluorescence image is rather crude method that would not allow us to draw robust conclusions. Instead, we have quantified the localization of MKs at the vessel walls – where perlecan is located – in WT and *G6b^-/-^* mice across the bone. We found no major difference between WT and KO mice.

**Author response image 1. respfig1:** Relative numbers of MKs located at the vessel.

The authors show differences in megakaryocyte size between WT and KO mice when cultured in the presence of different substrates. This difference could be due to different stages of maturation and differentiation, as the images suggest. Is there any difference in the maturation of megakaryocyte between phenotypes?

We previously showed that G6b-B-deficient MKs differentiate and proliferate normally ex vivo, under the conditions tested (Mazharian et al., 2012), which is now validated by EM analysis of bone marrow sections (Figure 9D-E). There is no evidence to suggest that differences in MK size observed in this experiment are due to developmental defects in the absence of G6b-B. Rather they are due to the ability of MKs to adhere and spread on different surfaces. We have incorporated this into the Discussion section.

7) In Figure 9C, number of cells quantified would again be good information to have in the figure legend.

MK numbers quantified have now been added to Figure 9C legend.

Axes should be added to aggregation graphs. Control (with no stimulation) should be shown. The data can be consolidated in a more palatable graph. Does plasma have an impact on HS binding to G6b-B?

Axes have been added to aggregation traces in the respective figure. Control, no stimulation, aggregation trace has also been added. We have tried presenting graphs in a more understandable format. We have yet to test the effects of plasma on HS binding by G6b-B.

8) The authors observe that perlecan does not seem to have an effect on either platelet aggregation (Figure 10) or G6b-B phosphorylation (Figure 12), likely due to the need for the proteoglycan to be immobilized on a surface, rather than in suspension. The effects of perlecan on platelet G6b-B signaling is a major component of this paper and should be elucidated by allowing platelets to adhere on immobilized perlecan prior to lysis and blotting for phosphotyrosine. The conclusion that G6b-B and its ligand perlecan inhibit activation in platelets is not really supported by direct evidence. The evidence is based on APAC binding to platelets, hence more direct evidence would be helpful.

The reviewer raises excellent points that we have tried to address. However, the issue is minimal platelet adhesion on perlecan, making it impossible to generate sufficient quantities of platelet lysates for Western blot analysis. We tried addressing this issue by incubating platelets on perlecan/fibrinogen co-coated surfaces, however, data was difficult to interpret due to strong G6b-B phosphorylation by the fibrinogen surface alone.

Furthermore, given the information in Figure 8A, platelets that receive an inhibitory stimulus via the HS/G6b-B interaction, would not be expected to adhere. We also did not observe an increase in G6b-B phosphorylation of the non-adherent fraction of platelets, most likely because only a small subset of platelets makes contact with the surface in a non-synchronized manner.

We respectfully disagree with the reviewer’s interpretation that there is no evidence for a role of perlecan regulating platelet function. Figure 8A,B clearly demonstrates an inhibitory effect of the HS side chains of perlecan on platelet adhesion and spreading. The inhibitory effect of perlecan on platelets was previously demonstrated by Whitelock and co-workers (Lord et al., 2009), but the mechanism of action was not known.

9) The text corresponding to Figure 11 and Figure 12 needs to be streamlined, as it is hard to follow. The figures could be edited/consolidated for clarification.

We agree with the reviewer and revised Figures and text. Figure 11 (now Figure 12) and corresponding supplements have been merged into a single figure. The two figure supplements for former Figure 12 (now Figure 13) have been combined into one figure, and redundant data has been removed.

10) Were data in Figure 11 obtained with washed platelets or platelets in plasma? Signaling is not induced by soluble perlecan. Can the authors immobilize perlecan to test if immobilized perlecan induces signaling?

The experiments were performed in mouse blood, diluted 1:10 in Tyrode’s buffer – we have specified this in the Figure legend.

Please see our answer to point 8 above regarding signalling experiment on immobilized perlecan.

11) In Figure 4—figure supplement 1, why do CHO cells transfected with G6b-B mutant bind less efficiently than mock transfected cells? One can suppose a dominant negative effect? CHO cells constitutively express G6b-B?

We thank the reviewer for raising these points. We have now reanalysed data using different settings and found a marginal overcompensation against the G6b-B channel in the data shown, giving rise to this difference. This has been corrected and data replaced from an experiment where this was not the case. CHO cells, do not express G6b-B; we gated on the entire population of mock-transfected cells for the histogram, using them as a negative control for G6b-B-transfected cells. This information has been added to the figure legend.

12) In Figure 8, given the important reduction in platelet binding onto fibrinogen and collagen coated surfaces in the presence of intact immobilized perlecan and given the percentage of adhesion, one would expect that, after removing HS from perlecan in the same settings, platelet adhesion to both ECMs should increase in an additive manner above the level of the two ECMs alone. This is further supported by the previous finding of Lord et al., 2011 showing that plaletet binding to C-terminal fragment of perlecan, endorepellin, enhances collagen-mediated platelet responses. Please comment.

Thank you for raising these points. We suspect this is due to saturating amounts of fibrinogen and perlecan, hence masking an additive effect.

13) The results shown in Figure 8B showed a negligible interaction of both WT and G6b-/- platelets to perlecan, this raises the doubt of the significance of the experiments shown in Figure 8C. Further, the differences in platelet area might be due to the intrinsic differences in platelet volume previously demonstrated by the same authors. Further, it is not clear how removing the HS and thus exposing the core protein of perlecan to platelet a2b1 platelet spreading should decrease.

We thank the reviewer for raising these points. Both we and others have shown that perlecan has an anti-adhesive property that we attribute to the HS-G6b-B interaction (Lord et al., 2009). Importantly, we demonstrate that this anti-adhesive property prevents platelet adhesion to fibrinogen and collagen. Hence, give that perlecan is the most abundant HS proteoglycan in the vessel wall, we disagree that these findings have no relevance.

Differences in spreading on perlecan are not due to the increase in volume of G6b KO platelets. This has been previously demonstrated (Mazharian et al., 2012), and new data has been added that supports this finding, including platelet spreading on fibrinogen, where no increase in volume of G6b KO platelets is observed (new Figure 8D). In addition, size differences between adhered G6b KO and WT platelets is abolished following treatment of perlecan with heparinase III. As mentioned in our answer to point 4, differences in size between WT and G6b KO platelets in the presence of HS points towards to existence of activating HS receptors on the platelet surface, masking the inhibitory effect of G6b-B. This is now clarified in the revised manuscript.

Of note, HS chains are attached to the N-terminus of the perlecan protein core, while endorepellin is found in the C-terminus, hence the α2β1 binding site should be accessible in the presence of the HS chains.

14) There is a lack of correspondence between the text and the panels shown in Figure 9. Although potentially interesting, the data on megakaryocytes appear too preliminary and do not add significance to the work.

We apologize again for the poor wording. We appreciate the reviewers comment and now have revised the figure, data and corresponding text and hope the reviewers find the new version easier to follow. Please also refer to our answer to points 5 and 6 above, regarding the significance of the MK data and the new data included.

15) The data on the opposite effects of soluble and conjugated heparin on platelet function are interesting. The authors concluded that these are due to a differential engagement of G6b-B and related downstream related signaling. To definitely prove the fundamental relevance of the SHPs-related inhibitory signaling, the authors should perform the experiments on platelet function with SHP1/2 KO platelets and by using SHPs specific inhibitors.

We currently do not have Shp1 and Shp2 single or double KO mice. Shp1/2 DKO mice have severe MK developmental defects, making it impossible to interpret functional findings. Although a specific Shp2 inhibitor is now available, a specific Shp1 inhibitor has yet to be developed. We would like to draw the reviewer’s attention to the data in our study using mice harbouring loss-of-function mutations in G6b-B. G6b-B diY/F is uncoupled from Shp1 and Shp2 in these mice (Geer et al., 2018), recapitulating the KO phenotype. Our findings clearly demonstrate a lack of an inhibitory effect of APAC on CLEC-2 mediated platelet activation (Figure 12 in the revised the manuscript).

Concerns about interpretation:1) While the authors demonstrate that perlecan is a ligand for G6b-B, their claim that this is the physiologically relevant ligand is not adequately supported. Given that prior work from mice and humans has demonstrated that the phenotype of reduced/absent G6b-B includes (1) decreased in vitro platelet aggregation/activation and (2) thrombocytopenia, it would seem logical to attempt to identify the ligand in lysates from tissues that would be relevant for those phenotypes. This would correspond to (1) blood/plasma (where platelets would be activated) or (2) bone marrow (where platelets are produced). Instead the authors looked for a G6b-B ligand in vena cava lysates; no rationale was provided as to the relevance of this to the human or mouse G6b-B knock out phenotypes. In fact, the authors note that their identified ligand perlecan would be inaccessible to platelets in intact blood vessels, making it hard to imagine that this would be a relevant ligand for platelet aggregation/activation.

We thank the reviewer for raising these points, however we respectfully disagree that the blood vessel wall, where perlecan is highly expressed, is not relevant to either platelet production or platelet activation. MKs routinely come into contact with sinusoidal blood vessels in the bone marrow and other sites of haematopoiesis, which we show express high levels of perlecan (Figure 9A). Although we have yet to definitely prove it, we hypothesize that the interaction between MK G6b-B and perlecan in sinusoidal blood vessels plays a critical role in regulating polarized proplatelet formation into the vessel lumen. Regarding platelet activation, many of the most potent and well characterized ligands of platelets are found in the vessel wall, including collagen, laminin and fibronectin that platelets only come into contact with following vascular injury. Moreover, the plasma protein VWF binds to exposed collagen and facilitates platelet tethering and adhesion to the site of injury. Thus, it follows that G6b-B, which regulates signalling from the respective receptors for these ligands is also present in the vessel wall, where it can regulate signalling strength from ITAM-containing receptors and integrins.

Although all blood vessels stained with recombinant G6b-B:Fc protein, we chose the vena cava to purify the ligand, as the signals was exceptionally strong in this vessel, and it is a large blood vessel that is easily isolated from negatively staining tissues.

These points are now included in the Results section and Discussion section of the revised manuscript.

However, also referring to point 1 on the heterogeneity of HS above, we recognize the possibility that G6b-B might also interact with other HS proteoglycans, which are located outside the vessel wall – we have incorporated this into the Discussion section.

2) Furthermore, additional information is needed to validate perlecan as a hit of the immunoprecipitation proteomics experiment. The hits from the G6b-B pulldown are listed in order of a "score" in Table 1, but it is not clear how the authors determined or defined this score. The authors mention that proteins that were "prominently present in the negative control" were not included in their list. It would be preferable if they could provide a complete data set for their proteomics results, and calculate a fold enrichment of the identified proteins/peptides in their experimental vs control condition. A reverse immunoprecipitation experiment (pulling down perlecan with an antibody and seeing if G6b-B is pulled down) would be a very useful experiment to confirm these findings.

Thank you for raising this point. We have updated Table 1 accordingly, which now depicts a list of all mass spectrometry hits, including those which were prominent in the negative control, most of which were myosins. The protein score was calculated using the SEQUEST HT search algorithm and is the sum of all peptide Xcorr values above the specified score threshold (0.8 + peptide_charge × peptide_relevance_factor where peptide_relevance_factor is a parameter with a default value of 0.4.) The score for each protein in the negative control is also depicted and the fold enrichment was calculated, where possible.

We have also provided the source data files of the experiments showing the complete data set. Please note that occasionally proteins show up in duplicate in this table, due to multiple entries in the Uniprot database and these duplicates were removed in Table 1. It should also be noted that Q3UHL6 – originally named *Putative uncharacterized protein* in the source data, was subsequently identified to be fibronectin. Table 1 was updated in light of this information.

We agree that a reverse immunoprecipitation would further validate the interaction between G6b-B and perlecan. However, we do not think this is necessary, because the interaction has already been demonstrated by multiple different techniques. Given the time required to establish the correct experimental conditions and with the limited additional value for the already extensive manuscript, we decided not to perform this experiment.

3) Further experimental detail is also needed in order to support the authors' claim that heparan sulfate and not chondroitin sulfate mediates binding of G6b-B to perlecan. The CRISPR knock out screen is suggestive, as it resulted in a number of hits in genes in the heparan sulfate biosynthesis pathway. However, only one hit from this screen was specifically validated by flow cytometry. It would be useful to validate a few more of the hits by making targeted CRISPR knock outs, and specifically to individually target EXTL3 and CSGALNACT_1/2_ (which are the enzymes that commit tetrasaccharide precursors to either heparan sulfate or chondroitin sulfate), and then look for G6b-B binding to the knock down cells by flow cytometry as in Figure 3C.

As the reviewers mentioned, our CRISPR data clearly shows that the binding is mediated by HS rather than CS. We have also presented additional evidence with biochemical experiments, showing that this interaction on the surface of the cells can be blocked by soluble heparin but not with the same concentration of soluble CS (Figure 3D). Given that we could corroborate the findings from the genetic screens with biochemical methods, we are confident that the interaction at the surface of cells is mediated by HS and not CS.

We have also performed binding of G6B to cells where *EXTL3* has been targeted (cells kept as polyclonal knockout lines) and have observed reduction in binding. As before, we have tested the binding of a HS-binding protein (APLP2, amyloid precursor-like protein 2) to ensure that binding is reduced in the *EXTL3*-targeted polyclonal lines.

**Author response image 2. respfig2:** A pentameric version of the G6b-B protein is used. CBFH refers to the COMP-β-lactamase-FLAG-3XHis tag. Binding assay was performed nine days post transduction with a sgRNA targeting *EXTL3* as described before (Sharma et al., *Genome Res*., 2018). Two separate batches of cells were infected with the same lentiviral preparation. Representative binding from one experiments is shown. APLP2 is a known HS-binding protein.

We would also like to share our unpublished work that points to G6b-B binding specifically to HS rather than CS. We have a protein that specifically binds to chondroitin sulphate and not heparan sulphate. We identified this binding behaviour from a genome-wide screen in HEL cell lines, in which we identified genes specific for CS (*CHST11*) biosynthesis and not for HS. We have a matching HEL cell line screen for G6b-B and we can clearly see the difference in the hits from these two screens. We see the presence of HS-specific genes (*EXTL3, EXT1, EXT2*) only for G6b-B and *CHST11* only for the CS-binding protein. Genes before the commitment step and *SLC35B2* are identified in both screens (Author response image 3).

**Author response image 3. respfig3:** Comparison of genome-wide screens performed for G6b-B (top panel) vs a CS-binding protein (bottom panel). G6B-BLH screened on HEL cells. Note the presence of HS-specific genes (*EXTL3, EXT1, EXT2*) only for G6B and *CHST11* only for the CS-binding protein.

When we tested this CS-binding protein on the same EXTL3 KO cell lines from Figure 1, we see that there is no loss in binding (Author response image 4).

**Author response image 4. respfig4:** Binding behaviour of a CS binding protein. Targeting *EXTL3* has no effect on binding whereas targeting *SLC35B2* clearly abrogates binding.

Given our experience with the binding behaviours of GAG binding proteins, we usually perform preliminary validation of all GAG binding proteins in *SLC35B2*-KO lines and perform secondary validation with biochemical approaches.

It would also be helpful to generate a perlecan CRISPR HEK293 knock out and test for G6b-B binding by flow.

We did not specifically identify HSPG2 in the genetic screen and we believe that the binding of G6B on the surface of transformed cell lines is mediated by HS-side chains in *general* rather than HS-side chains on a *particular* protein backbone. A number of cell surface HSPGs can function equivalently on the cell line to provide HS chains and this would preclude the identification of one specific ligand. While the genetic screen itself reveals the HS-binding properties of G6B, it is not well-suited to identify the proteoglycan backbone itself.

4) Finally, the functional data that are provided here lead the authors to conclude that the function of G6b-B is to inhibit platelet activation in the presence of perlecan heparan sulfate. Given that the phenotype of decreased/absent G6b-B is bleeding and decreased platelet activation, it is surprising that the in vitro experiments suggest that absence of G6b-B would result in excessive platelet activation. This seeming contradiction calls in to question whether perlecan heparan sulfate is truly the physiologically pertinent ligand for the clinical phenotype.

We have previously shown that the bleeding phenotype of G6b KO mice is due to a severe reduction in platelet count and down-regulation of the ITAM-containing collagen activation receptor GPVI-FcR γ-chain (Mazharian et al., 2012). G6b null mutations resulted in a similar phenotype in humans (Hofmann et al., 2018). Uncoupling of G6b-B from the tyrosine phosphatases Shp1 and Shp2 also resulted in a similar phenotype in mice (Geer et al., 2018). The reduction in platelet count is primarily due to a reduction in proplatelet production by G6b-B-deficient megakaryocytes. There is an accompanying reduction in platelet half-life in G6b KO mice that likely contributes to the thrombocytopenia. This is due to G6b KO platelets being in a ‘pre-activated’ state, and being recognized as such by resident macrophages in the spleen and liver, and more rapidly cleared from the circulation compared with control platelets.

The severe reduction in GPVI-FcR γ-chain in platelets from G6b KO and loss-of-function mouse models is a negative feedback mechanism due to tonic signalling from GPVI-FcR γ-chain in the absence of G6bB and collagen. Consequently, platelets from these mouse models do not become activated or adhere normally to collagen, contributing to the bleeding diathesis in these mice (Mazharian et al., 2012). Further evidence of the inhibitory function of G6b-B is hyper-activation of G6b KO platelets to CLEC-2 agonists (Mazharian et al., 2012), and inhibition of GPVI and CLEC-2 signalling by G6b-B in transiently transfected DT40 cells (Mori et al., 2008).

Together with biochemical and structural data presented in our manuscript, we believe this provides compelling evidence that the HS chains of perlecan are indeed ligands of G6b-B. We hypothesize this interaction is critical for regulating proplatelet formation when MKs come into contact with perlecan in sinusoidal blood vessels in the bone marrow, and the platelet response to vascular injury by moderating signalling from ITAM-containing and integrin receptors. Our study lays the foundation for further studies of the physiological relevance of this interaction in mice and humans.